# TOKEN-LEVEL GUIDED DISCRETE DIFFUSION FOR MEMBRANE PROTEIN DESIGN

## ABSTRACT

Reparameterized diffusion models (RDMs) have recently matched autoregressive methods in protein generation, motivating their use for challenging tasks such as designing membrane proteins, which possess interleaved soluble and transmembrane (TM) regions. We introduce the *Membrane Diffusion Language Model* (**MemDLM**), a fine-tuned RDM-based protein language model that enables controllable membrane protein sequence design. MemDLM-generated sequences recapitulate the TM residue density and structural features of natural membrane proteins, achieving comparable biological plausibility and outperforming state-of-the-art diffusion baselines in motif scaffolding tasks by producing lower perplexity, higher BLOSUM-62 scores, and improved pLDDT confidence. To enhance controllability, we develop **Pe**r-**T**oken **G**uidance (**PET**), a novel classifier-guided sampling strategy that selectively solubilizes residues while preserving conserved TM domains, yielding sequences with reduced TM density but intact functional cores. Importantly, MemDLM designs validated in TOXCAT $\beta$-lactamase growth assays demonstrate successful TM insertion, distinguishing high-quality generated sequences from poor ones. Together, our framework establishes the first experimentally-validated diffusion-based model for rational membrane protein generation, integrating *de novo* design, motif scaffolding, and targeted property optimization.

## 1 INTRODUCTION

Membrane proteins play a crucial role in biological systems, regulating molecular transport, signal transduction, and cellular communication (Jelokhani-Niaraki, 2022). Their capacity to bind specific ligands or undergo conformational changes renders them essential targets for drug development and therapeutics for various diseases (Sanganna Gari et al., 2021). Even more interestingly, *de novo* design and engineering of membrane proteins offers a powerful therapeutic modality by enabling the creation of highly-specific and stable proteins that can precisely modulate cell signaling pathways, transport processes, and immune responses, making them ideal for targeting diseases such as cancer and neurological disorders (Jelokhani-Niaraki, 2022). Current methods for designing new protein sequences or scaffolds rely on pre-trained structure prediction networks (Wang et al., 2022; Yin et al., 2007; Elazar et al., 2022), which remains a particularly challenging prerequisite for membrane protein targets. The scarcity of high-resolution structures hinders the training of high-fidelity deep learning structure prediction models for membrane proteins: only $\sim 1\%$ of the current PDB structures are annotated as membrane proteins. Further, energy functions underlying physics-based computational models are suboptimal because they often require iterative optimizations to design analogs of membrane proteins (Vorobieva et al., 2021). As a result, current methods in *de novo* membrane protein design are limited to simple helical barrel or beta-barrel folds with low sequence complexity.

While deep learning-based topology predictors (e.g., DeepLoc, AllesTM) aid in identifying helix regions and subcellular localization, they primarily analyze existing sequences and do not support *de novo* generation for function-specific design (Thumuluri et al., 2022) (Hönigschmid et al., 2020). Prior computational design efforts have achieved impressive results by designing zinc-transporting helices, yet they are often limited to fixed scaffolds, small proteins, or require extensive intervention (Joh et al., 2014). What remains missing is a generative modeling framework that can autonomously produce membrane protein sequences with controllable structural features, including TM helices,

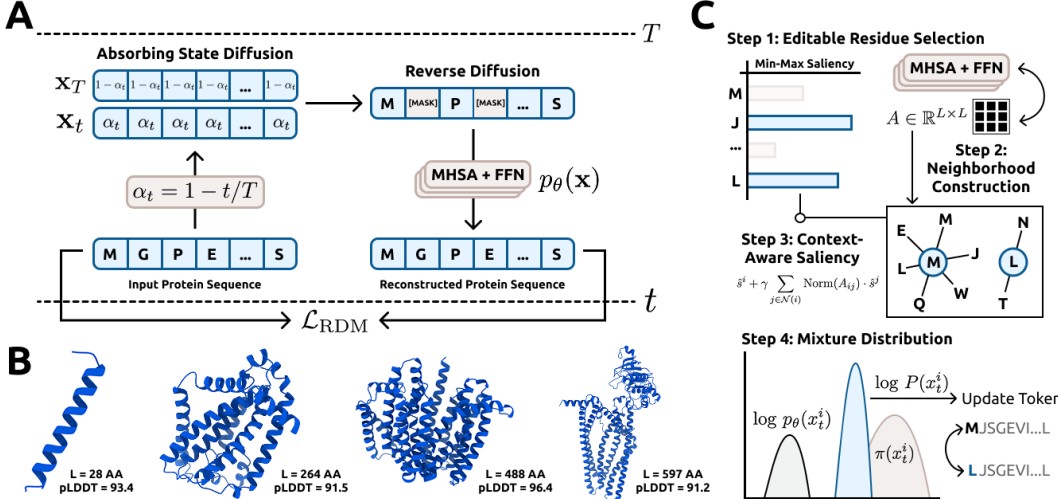

Figure 1: **MemDLM Schematic**. **A)** RDM-based model training diagram. **B)** AlphaFold3 visualizations of unconditional samples. **C)** Token-level classifier guided diffusion sampling with PET algorithm.

soluble domains, and higher-order topologies, without relying on predetermined scaffolds or manual adjustments (Goverde et al., 2024).

Discrete diffusion models have recently emerged as powerful tools for generative modeling in biological sequence spaces, including proteins, peptides, and nucleic acids (Austin et al., 2021; Sahoo et al., 2024; Wang et al., 2024; Shi et al., 2024; Peng et al., 2025; Tang et al., 2025a;b; Nisonoff et al., 2025; Rector-Brooks et al., 2025). These models operate by progressively denoising masked inputs, allowing them to capture long-range dependencies without requiring autoregressive factorization. However, while discrete diffusion excels at unconstrained generation, guiding these models toward property-specific objectives still remains an unsolved challenge (Schiff et al., 2025). Existing classifier-based and classifier-free guidance methods often struggle to enforce token-level constraints, suffer from noisy gradient estimates, or fail to preserve structural elements essential to biological function (Nisonoff et al., 2025; Rector-Brooks et al., 2025; Wang et al., 2024). In the context of membrane protein design, where transmembrane (TM) domains must be preserved even during optimization, these limitations make existing guidance strategies insufficient.

In this work, we introduce **MemDLM**, a discrete diffusion protein language model for rational membrane protein design (Figure 1). At the core of our approach is *PEr-Token Guidance* (PET), a novel classifier-guided sampling algorithm that combines attention scores and classifier rewards to optimize specific sequence tokens during inference. Unlike traditional classifier-guidance methods (Gruver et al., 2024; Li et al., 2024; Vignac et al., 2022; Dhariwal & Nichol, 2021; Tang et al., 2025a; Chen et al., 2025; Schiff et al., 2025), PET ensures the retention of targeted tokens, an essential requirement in membrane protein design, where highly conserved transmembrane (TM) domains are critical to maintaining structural topology. We demonstrate that MemDLM generates biologically relevant proteins with structural features resembling membrane proteins (e.g. $\alpha$-helical bundles) and show that PET solubilizes natural membrane proteins while retaining key functional TM domains. Overall, our integrated pipeline serves as a versatile, end-to-end platform for designing and optimizing membrane protein sequences, with potential applications spanning therapeutics, drug delivery, and synthetic biology.

**Our key contributions are as follows:**

- We introduce **MemDLM**, a discrete diffusion protein language model specifically fine-tuned for *de novo* generation of membrane protein sequences with controllable structural features.

- We develop PET, a novel classifier-guided sampling algorithm to optimize specific sequence tokens during inference, ensuring the retention of targeted amino acid tokens like conserved TM domains.

- We demonstrate that MemDLM enables controllable sequence generation through token-level editing. In practice, we show MemDLM effectively solubilizes existing natural membrane protein sequences while preserving crucial functional TM regions.

- We motivate MemDLM's utility in real-world therapeutic design by showing it (i) outperforms existing state-of-the-art models by achieving improved sequence-specific computational benchmarks in *de novo* generation and sequence scaffolding tasks, and (ii) produces experimentally-validated membrane protein designs that exhibit favorable growth curves under antibiotic selection.

## 2 METHODS

**Language Modeling Preliminaries** Let $\mathbf{x} = (x^1, x^2, \ldots, x^L) \in \{0, 1\}^{L \times |\mathcal{V}|}$ denote a discrete sequence of length $L$, where each token is represented as a one-hot vector over the vocabulary $\mathcal{V} = \{0, 1, \ldots, 32\}$. The vocabulary includes 25 canonical and non-canonical amino acids, along with several special tokens (Lin et al., 2023). *Language modeling* aims to estimate the underlying data distribution $\mathbf{x} \sim q(\mathbf{x})$ using a parameterized probabilistic model $p_\theta(\mathbf{x})$. Since the true distribution $q(\mathbf{x})$ is typically intractable, we approximate it using a neural network with parameters $\theta$. In Supplementary A.1, we lay out the foundation for RDM-based protein language models by considering related modeling paradigms.

### 2.1 MEMDLM

**Modeling** MemDLM is built on the Reparameterized Diffusion Model (RDM) framework (Zheng et al., 2023). We define $\text{CAT}(x; \mathbf{p})$ as the categorcial distribution on the discrete sequence $\mathbf{x}$ governed by the vector $\mathbf{p} \in \Delta^{|\mathcal{V}|-1}$, where $\Delta^{|\mathcal{V}|-1}$ denotes the $(|\mathcal{V}| - 1)$-dimensional probability simplex. Given a stationary noise distribution $\mathbf{q}_{\text{noise}}$, we define the unconditional prior as $q(\mathbf{x}_t) = \prod_{i=1}^{L} \text{CAT}(x_t^i; \mathbf{q}_{\text{noise}})$. We can then write the *forward* diffusion process as a transition kernel defined in closed-form as a convex combination of clean data and noise:

$$q(\mathbf{x}_t | \mathbf{x}_{t-1}) = \alpha_t \mathbf{x}_0 + (1 - \alpha_t) q_{noise} \tag{1}$$

where $\alpha_t = \prod_{i=1}^{t} \beta_i = 1 - t/T$ is a linear noise schedule. This transition distribution in Eq. 1 shows that the forward process is ultimately a convex combination of $\alpha_t$, the probability of clean data $\mathbf{x}_0$ remaining unchanged, and $1 - \alpha_t$, the probability of $\mathbf{x}_0$ transitioning to the [MASK] token. By sampling $t \sim \mathcal{U}(0, T = 500)$, we can determine the identity of a token at the given timestep of the forward process:

$$x_t^i = \begin{cases} [\text{MASK}] & \text{if } u_i < \frac{t}{T}, \quad u_i \sim \text{Uniform}(0, 1) \\ x_0^i & \text{otherwise} \end{cases} \tag{2}$$

Importantly, the forward noising process is characterized by an *absorbing state*: $\lim_{t \to T} \alpha_t = \lim_{t \to T}(1 - t/T) = 0$, indicating all tokens are guaranteed to be replaced by noise. During inference, $\text{MemDLM}_\theta$ must *denoise* a fully masked sequence $\mathbf{x}_T = \{[\text{MASK}]\}_{i=1}^{L}$, rendering the absorbing state a necessary ingredient of the forward noising process. In Section 2.2, we formally outline a generalized denoising framework from Peng et al. (2025) to obtain samples from masked diffusion models (e.g., RDMs).

**Loss Function** Following the proof in Wang et al. (2024) (Appendix Section A), the RDM framework simplifies the ELBO (Eq. 12) by breaking down the KL-divergence term to yield a simplified training objective:

$$\mathcal{L}_{\text{RDM}} = -\mathbb{E}_{q(\mathbf{x}_0)} \text{KL} \left[ q(\mathbf{x}_{t-1} \mid \mathbf{x}_t, \mathbf{x}_0) \parallel p_\theta(\mathbf{x}_{t-1} \mid \mathbf{x}_t) \right]$$

$$= \mathbb{E}_{q(\mathbf{x}_0)} \left[ \lambda_t \sum_{i=1}^{L} b^i(t) \cdot \log p_\theta(x_0^i \mid \mathbf{x}_t) \right] \tag{3}$$

where $\lambda_t := T - (t - 1)$ represents a linear, time-dependent coefficient and $b^i(t) := \mathbf{1}_{x_t^i \neq x_0^i}$ . In practice, $\mathcal{L}_{\text{RDM}}$ can easily be computed using the cross-entropy loss between logits and sequence labels. In Supplementary B.2, we detail the specific architectural and training schemes used to construct MemDLM.

## 2.2 Path-Planning Sampling

To generate realistic membrane-like protein sequences from a trained MemDLM, we adopt the Path-Planning (P2) paradigm introduced by (Peng et al., 2025), a novel sampling framework for masked discrete diffusion language models. Notably, P2 breaks the assumption of uniform unmasking probabilities and enhances generative quality compared to stochastic sampling from a Gumbel-Softmax distribution or greedy decoding of softmax logits. We follow the *self-planner* variant of P2, where the denoiser itself provides a planning signal used to identify and resample low-value tokens. Here and in Algorithm 2, we outline the key steps of self-planning in P2 but direct the reader to (Peng et al., 2025) for a complete background.

**Initial Token Sampling** Beginning with a fully masked sequence $\mathbf{x}_t = \{[\text{MASK}]\}_{i=0}^L$, MemDLM predicts denoised logits $\mathbf{z}_{t-1} \in \mathbb{R}^{L \times |\mathcal{V}|}$ via $\mathbf{z}_{t-1} = p_\theta(\mathbf{x}_t)$ at each timestep. Candidate tokens are sampled from the logits using Gumbel-softmax decoding with temperature parameter $\tau$:

$$x_{t-1}^i = \arg\max_v \left( \log \text{softmax} \left( \frac{z_{t-1}^{i,v}}{\tau} + g^{i,v} \right) \right), \quad \mathbf{g}_i \sim \text{Gumbel}(0, 1) \tag{4}$$

**Self-Planning** An important requirement of self-planning is resampling low-value tokens using the predictions of the denoising model. Accordingly, we use MemDLM's log probabilities to compute $s_t^i = \log p_\theta(x_t^i)$, a per-position score, and $\mathcal{R}_t = \mathbf{x}_{t-1}^{\setminus \mathcal{M}}$, the set of unmasked positions $\setminus \mathcal{M}$ eligible for remasking. We select the top-$K$ tokens from $\mathcal{R}_t$ with the lowest log-probability scores $s_t^i$ and remask them. Specifically, we dynamically compute $K = \lfloor (1 - \kappa_t) \cdot |\mathcal{R}_t| \rfloor$ as a fixed proportion of unmasked positions controlled by the monotonic scheduling function $\kappa_t = \kappa(i/N)$, where $i \in \{1, 2, \ldots, N\}$ and $\kappa : [0, 1] \rightarrow [0, 1]$. This update forces the token predictions MemDLM was not confident about (low $s_t^i$) to be remasked.

**Token Resampling** We sample new tokens at the remasked positions by copying the most recent denoised tokens from the previous timestep $\mathbf{x}_{t-1}$ into the current sequence $\mathbf{x}_t$ at positions that were masked but are no longer among the $K$ lowest-scoring tokens. This step progressively commits high-confidence tokens while leaving low-confidence regions available for further refinement in future steps, a key advantage over ancestral and greedy sampling schemes. By following the self-planning scheme of P2, no additional model training or overhead is required, providing a lightweight inference mechanism for membrane protein design tasks.

## 2.3 Per-Token Classifier Guided Sampling

While generating arbitrary membrane proteins is valuable, it is insufficient for downstream applications, as unconditional samples are unlikely to exhibit the functional properties required for their use as therapeutic modalities (Jelokhani-Niaraki, 2022). *Classifier-guided sampling* has recently introduced controllability to deep generative models by following a gradient signal from a pre-trained classifier model (Gruver et al., 2024), (Li et al., 2024), (Vignac et al., 2022), (Dhariwal & Nichol, 2021), (Tang et al., 2025a), (Chen et al., 2025). Although these methods bias the model's sampling trajectory towards the desired class label, there is no guarantee that specific sequence tokens are preserved during inference. Most similar to our work is LaMBO-2 (Gruver et al., 2024), a classifier-guidance mechanism for discrete diffusion models. In Supplementary A.2, we present a rigorous dissection of the method to motivate the need for a classifier-guidance strategy that can preserve an initial sequence scaffold.

To this end, we introduce **Per-Token Guidance** (PET), a novel classifier-guided sampling algorithm that selects and replaces specific sequence tokens with optimized analogues, moving the overall sequence towards the desired property (Figure 1C, Algorithm 3). In the case of membrane protein

design, PET can readily be used to replace noncritical TM residues with soluble analogues to guarantee overall sequence solubility while maintaining biologically conserved TM domains. Solubilizing membrane proteins without disrupting these critical TM residues is essential for ensuring functional foldability and membrane localization, as TM residues often mediate key structural and biophysical interactions. Below, we carefully outline our PET algorithm and refer the reader to Supplementary A.2 for a background on discrete classifier guidance.

**Setup** Given a sequence consisting of only amino acid tokens, $\mathbf{x} = \{x_i \in \text{Canonical}\}_{i=1}^L$, PET first identifies a dynamic subset of editable positions $\mathcal{E} \subseteq \{1, \ldots, L\}$ using existing residue annotations or a trained per-token solubility classifier $v_\phi : \mathbb{R}^{B \times L \times D} \to \mathbb{R}^{B \times L}$. This classifier operates over the hidden states $h$ derived from the ESM-2-650M protein language model (Lin et al., 2023) and is trained on fully unmasked sequences. See Section B.3 for full training details regarding $v_\phi$.

**Determining Editable Residues** PET first constructs a set of conserved, *non-editable* token indices $\mathcal{C}$ based on solubility annotations or predictions:

1. If soluble residue annotations $\mathcal{S} \subseteq \{1, 2, \ldots, L\}$ are provided (e.g. experimentally-derived labels for known membrane protein sequences), initialize $\mathcal{C} = \mathcal{S}$.

2. If no annotations are provided, initialize $\mathcal{C} = \{i \in \{1, \ldots, L\} \mid v_\phi(h_t)_i \geq 0.5\}$. Inherently, it is assumed that some $v_\phi(h_t)_i < 0.5$.

Next, PET identifies additional tokens to add to the conserved, non-editable set $\mathcal{C}$. Specifically, we consider tokens with *low-editability – i.e.*, residues predicted to be *insoluble*, which we use as a proxy for transmembrane (TM)-like character. It is critical to preserve the most conserved TM regions during optimization in order to maintain the biological plausibility of the generated membrane protein. To that end, PET guides the selection of these residues using LaMBO-2's (Supplementary Section A.2) definition of a token's *saliency* $s^i(h)$, a score that quantifies the importance of token $i$ relative to the classifier $v_\phi$ (Gruver et al., 2024). Given a sequence's latent representation, we construct a *saliency map* $\mathbf{s} = (s^1, s^2, \ldots, s^L) \in \mathbb{R}^L$:

$$\mathbf{s}(h) := \max\left\{ \left( \sum_{d=1}^D |\nabla_h v_\phi(h)_d| \right)^{1/\tau}, \epsilon \right\}, \quad \hat{s}^i := \frac{s^i - \min \mathbf{s}}{\max \mathbf{s} - \min \mathbf{s} + \delta} \tag{5}$$

using temperature $\tau = 2.0$ and a gradient noise floor $\epsilon = e^{-4}$ to stabilize gradient noise. Although LaMBO-2 normalizes the saliency map to a probability distribution $P_{\text{edit}}(\mathbf{x}_t) = \mathbf{s}/\sum_i s_i$ (Gruver et al., 2024), PET opts for min-max scaling (Eq. 5) to prevent vanishing probabilities when $L$ is large. If $v_\phi$ is well-trained, high values of $\mathbf{s}$ should correlate with structurally critical (i.e., low-editability) TM-like residues. To finalize the set of conserved tokens, PET selects the top-$K$ most salient tokens from the remaining non-soluble residues, where:

$$\mathcal{C} = \mathcal{C} \cup \text{top-}K\left(\hat{\mathbf{s}}, K = \max\left\{1, \tfrac{1}{10} \cdot (L - |\mathcal{C}|)\right\}\right), \quad \mathcal{E} = \{1, \ldots, L\} \setminus \mathcal{C} \tag{6}$$

Together, these token selection strategies define $\mathcal{E}$, the set of editable token indices. This set excludes soluble and highly salient residues to preserve membrane protein character (TM-like residues) while optimizing for sequence solubility.

**Neighborhood Construction.** For each editable token $i \in \mathcal{E}$, PET constructs a context-aware *neighborhood* $\mathcal{N}(i)$ based on attention scores. Let $A \in \mathbb{R}^{L \times L}$ be the attention matrix extracted from the final Transformer layer of $p_\theta$. The neighborhood $\mathcal{N}(i)$ is formed using top-$p$ nucleus sampling over the normalized attention weights $\text{Norm}(A_{i,:}/\tau)$, excluding special tokens and the self-position $i$; we set $\tau = 1/\log L$ to ensure neighborhood selection is neither overly diffuse in long sequences nor overly narrow in short sequences. Thus, the final neighborhood contains all tokens $j$ such that the cumulative attention probability $\sum_{j' \in \mathcal{N}(i)} A_{ij'}$ exceeds the threshold $p = 0.9$. The construction of an attention-informed neighborhood is necessary to propagate long-range residue information to avoid blindly modifying individual tokens.

**Context-Aware Saliency**   PET then refines a token's raw saliency score $s_i$ with contributions from the token's attention-weighted neighborhood $\mathcal{N}(i)$. The *context-aware saliency* score $\tilde{s}^i$ is defined as:

$$\tilde{s}^i := \hat{s}^i + \gamma \sum_{j \in \mathcal{N}(i)} \frac{A_{ij}}{\sum_{j' \in \mathcal{N}(i)} A_{ij'}} \cdot \hat{s}^j \tag{7}$$

where $\gamma = 0.5$ controls the influence of the neighborhood saliency. Overall, the context-aware saliency blends both the intrinsic importance of the token $x^i$ with the contributions of tokens it attends to most strongly, creating a holistic representation of an individual residue's contribution to sequence-level solubility.

**Mixture Distribution**   Let $\log p_\theta(x_t^i)$ be the log-probability distribution across the vocabulary for a singular token by the language model at timestep $t$, and let $\pi(x_t^i)$ be a prior token distribution in log-space. To update a token, PET defines a *mixture distribution* $\log P(x_t^i)$ for each editable position $i \in \mathcal{E}$:

$$\log P(x_t^i) = (1 - w^i) \cdot \log p_\theta(x_t^i) + w^i \cdot \pi(x_t^i) \tag{8}$$

By construction, $P(x_t^i)$ remains a valid probability distribution, as it is a convex combination of two normalized distributions. The mixture weight $w^i$ can be computed as:

$$w_i = \sigma(\alpha \cdot \tilde{s}^i) \tag{9}$$

with $\sigma(\cdot)$ denoting the sigmoid function and $\alpha = 5.0$ controlling the sharpness of the transition. Eq. 8 ensures that an updated token's distribution is biased towards the prior when $\tilde{s}^i$ is large since $s_i \to 1$ when $v_\theta(h_t^i) \to 0$. Biologically, this corresponds to a residue with high TM-like character that is thus conserved and should remain fixed. Conversely, when $\tilde{s}^i$ is small, PET favors the model's default prediction, allowing more flexibility in low-saliency (non-critical) positions.

**Prior Distribution**   In order to consutrct the mixture distribution, we define a *temporal prior* $\pi(x_t^i) := \log p_\theta(x_{t-1}^i)$ in PET sampling that leverages the denoising model's log probabilities from a previous diffusion timestep. This formulation maintains the likelihood of the original sequence while encouraging updates from the mixture weighting in Eq. 8.

**Token Sampling and Preservation.**   A new token $\hat{x}^i$ is sampled from $P(x^i)$ for each position $i \in \mathcal{E}$. By design, PET will not update positions $j \notin \mathcal{E}$, resulting in an optimized sequence that preserves soluble and conserved TM regions while refining low-saliency, TM positions. To produce optimized amino acid tokens, we sample from a categorical distribution parameterized by the updated token probabilities at each position, $\hat{x}^i \sim \text{CAT}(\log P(x^i))$.

## 2.4   TOXCAT-$\beta$-LACTAMASE GROWTH ASSAY

The TOXCAT-$\beta$-lactamase assay was used to evaluate membrane insertion and TM association of MemDLM-generated sequences (Russ & Engelman, 1999; Lis & Blumenthal, 2006). Candidate designs were cloned between an N-terminal ToxR transcriptional activator and a C-terminal periplasmic $\beta$-lactamase in the pMAL_dst$\beta$L vector, and transformed into *E. coli* Cloni cells. Single colonies were used to inoculate LB cultures with spectinomycin (50 µg/mL), diluted to $OD_{600} = 0.05$, and normalized to $1.95 \times 10^5$ cells per well in 96-well plates. The cells were subjected to different selective pressures: carbenicillin (300 µg/mL) to report on membrane insertion, or combined carbenicillin (100 µg/mL) and chloramphenicol (100–120 µg/mL) to report on TM-mediated oligomerization. Plates were incubated at $37°$C with continuous shaking in a BioTek Synergy H1 plate reader, and growth was monitored by $OD_{600}$ every 10 minutes for 24 hours. Successful insertion positions $\beta$-lactamase in the periplasm to hydrolyze carbenicillin, while oligomerization activates the *ctx* promoter via ToxR dimerization, conferring chloramphenicol resistance.

## 3 EXPERIMENTS

### 3.1 *De Novo* GENERATION

**Setup** Unconditional generation of natural membrane protein sequences expands the landscape of rational protein design. To this end, we use MemDLM to *de novo* generate 1,098 membrane protein sequences, setting the sequence length based on the distribution of the test set. We opted to use the test set's length distribution as the basis for our experiments to yield a realistic evaluation of sequence plausability and membrane character. We benchmarked MemDLM against the Diffusion Protein Language Model (DPLM, (Wang et al., 2024)) and ProGen2 Nijkamp et al. (2023) to validate MemDLM's generative quality relative to SOTA models (Supplementary Section B.4). To generate sequences, we use ancestral sampling for ProGen2 to align with its autoregressive training regime and P2 Self-Planning for MemDLM and DPLM. Finally, given the limited availability of experimentally-verified membrane structures, we focused on sequence-based metrics (Supplementary Section B.5). Notably, we computed the TM Residue Density of the generated sequences by predicting TM and soluble residue regions with DeepTMHMM (Hallgren et al., 2022).

It is also worth noting that ProGen2 presents several limitations in this setting. First, its autoregressive design restricts it to ancestral sampling, preventing the model from performing sequence infilling tasks. Second, because ProGen2 relies on a byte-pair encoding (BPE) tokenizer, it cannot guarantee exact sequence lengths during generation. Finally, a substantial fraction of ProGen2 outputs (366 of 1,098 sequences) were excluded from evaluation because they contained non-canonical tokens such as numbers or unnatural amino acids. For these reasons, we restrict our use of ProGen2 to unconditional sequence generation tasks.

Table 1: Computational validation of generated and experimentally-validated membrane protein sequences. The performance of MemDLM is compared against SOTA discrete protein generative models, including ProGen2 and DPLM. TMRD denotes TM Residue Density

|  | pLDDT ($\uparrow$) | TMRD | PPL ($\downarrow$) | ENTROPY ($\uparrow$) |
|---|---|---|---|---|
| Test Set | $76.637_{\pm 10.676}$ | $0.294_{\pm 0.219}$ | $5.707_{\pm 3.435}$ | $3.918_{\pm 0.253}$ |
| ProGen2 | $54.998_{\pm 19.235}$ | $0.048_{\pm 0.153}$ | $126.646_{\pm 1415.166}$ | $2.622_{\pm 1.290}$ |
| DPLM | $62.318_{\pm 20.669}$ | $0.310_{\pm 0.264}$ | $\mathbf{6.323}_{\pm 10.317}$ | $3.179_{\pm 0.812}$ |
| MemDLM | $\mathbf{67.410}_{\pm 14.828}$ | $\mathbf{0.311}_{\pm 0.250}$ | $6.344_{\pm 3.278}$ | $\mathbf{3.743}_{\pm 0.326}$ |

The results show that MemDLM generates sequences with a solubility profile (TM Residue Density) that closely matches that of experimentally-verified membrane proteins, indicating that MemDLM has successfully learned their underlying distribution. Further, MemDLM-generated sequences are more likely to fold into biologically plausible structures compared to DPLM and ProGen2, evidenced by MemDLM's higher pLDDT scores. Although DPLM and MemDLM achieve similar ESM-2-650M Pseudo Perplexities, DPLM's low Shannon entropy metric suggests that DPLM generates more repetitive amino acid sequences. Overall, these results suggest that MemDLM has more effectively captured the underlying distribution of membrane protein sequences through an RDM-based training strategy (Supplementary A1). As a final validation, we visualize MemDLM-generated sequences with AlphaFold3 (Supplementary D) and confirm the presence of hallmark membrane protein structures, including $\alpha$-helical bundles and distinct TM and soluble regions (Zhang et al., 2015).

### 3.2 EXPERIMENTAL VALIDATION

**Setup** To fully validate MemDLM's *de novo* generative capabilities, we selected three generated sequences considered to be *single-pass* membrane proteins ("GoodTM") from the top-100 and two from the bottom-22 ("PoorTM") set of MemDLM-generated sequences for experimental validation in *Escherichia coli* (*E. coli*) using TOXCAT-$\beta$-lactamase bacterial growth assays (Lis & Blumenthal, 2006), which employ a dual-reporter system for evaluating membrane insertion and oligomerization of single-pass peptides and proteins (Russ & Engelman, 1999; Lis & Blumenthal, 2006; Ottemann & Mekalanos, 1995; Armstrong & Senes, 2016; Elazar et al., 2016) (Supplementary Sections B.6, C.3). In these constructs, the design of interest is inserted between an N-terminal ToxR cytoplasmic domain and a C-terminal periplasmic $\beta$-lactamase. *E. coli* survival under different antibiotic selection

pressures then provides a direct functional readout: survival in carbenicillin indicates successful membrane insertion, which positions the $\beta$-lactamase in the periplasm to degrade the antibiotic, while growth in carbenicillin and chloramphenicol demonstrates TM-mediated oligomerization, where multimerization of the ToxR transcription factors activates the downstream ctx promoter that confers resistance to chloramphenicol.

**Results** Figure 2 shows the TOXCAT growth curves for poor and high-quality MemDLM sequences alongside the positive insertion controls GpA and CLS (Supplementary Figure A5). Under carbenicillin selection (300 $\mu$g/mL), GpA, CLS, GoodTM4, GoodTM5, and GoodTM8 all achieved similar growth kinetics and reached the midpoint of log-phase growth at $\sim$4 hours, demonstrating similar membrane insertion efficiencies. PoorTM4 showed no growth in carbenicillin, much like our negative controls (Supplementary Figure A6), indicating that the sequence is not membrane-inserting. However, PoorTM2, which contains six charged residues within the predicted TM span, also

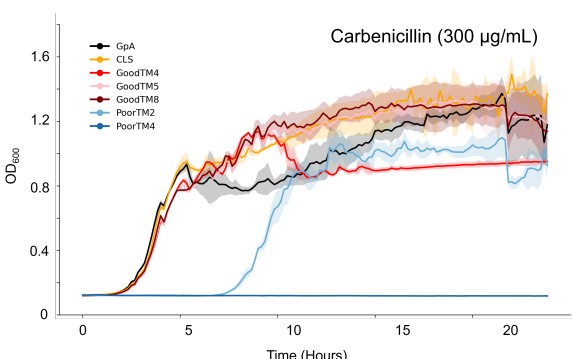

Figure 2: Growth curves of MemDLM-generated TM sequences under carbenicillin (300 $\mu$g/mL).

grew in carbenicillin but with a noticeable delay, suggesting weaker membrane insertion propensity. The survival of GoodTM designs under carbenicillin selection demonstrates that MemDLM can generate *de novo* TM-inserting sequences and that filtering generated sequences with computational metrics effectively ranks TM-like sequences. The poor survivability of PoorTM2 and PoorTM4, both ranked among the bottom 22 sequences by MemDLM, compared to the GoodTM designs further supports MemDLM's ability to distinguish TM-like sequences.

### 3.3 MOTIF SCAFFOLDING

**Setup** As a natural extension of *de novo* design, we scaffolded around TM and soluble motifs of experimentally-annotated membrane proteins. We take the entire test set, comprising 1,098 experimentally-verified membrane protein sequences with annotated TM and soluble motifs, and mask out all residues except those in the TM or soluble motif(s). We use these partially masked sequences as input to the models to assay their capability to generate scaffolds conditioned on known TM or soluble motifs. We focused on these domains due to their distinct hydrophilic and hydrophobic regions that govern the folding and thus function of the overall protein. Like unconditional generation, our evaluations focus on comparing MemDLM's performance against SOTA discrete diffusion protein models, namely EvoDiff (Alamdari et al., 2023).

**Results** Our results (Table 2, Supplementary Figures A2, A3) show that MemDLM scaffolds functional motifs with greater confidence while preserving biologically critical regions compared to SOTA discrete diffusion models.

Table 2: Reconstruction quality comparison of models scaffolding around TM and soluble motifs of 1,098 experimental membrane protein sequences that represent the MemDLM model test set.

| | MOTIF | PLDDT ($\uparrow$) | PPL ($\downarrow$) | BLOSUM ($\uparrow$) | ENTROPY ($\uparrow$) |
|---|---|---|---|---|---|
| Test Set | Insol | $76.637_{\pm 10.676}$ | $5.707_{\pm 3.435}$ | – | $3.918_{\pm 0.253}$ |
| | Sol | $76.637_{\pm 10.676}$ | $5.707_{\pm 3.435}$ | – | $3.918_{\pm 0.253}$ |
| EvoDiff | Insol | $\mathbf{64.058}_{\pm 19.229}$ | $9.841_{\pm 4.091}$ | $2.176_{\pm 1.587}$ | $3.841_{\pm 0.268}$ |
| | Sol | $64.036_{\pm 19.145}$ | $4.632_{\pm 3.271}$ | $-0.188_{\pm 1.134}$ | $\mathbf{3.841}_{\pm 0.268}$ |
| MemDLM | Insol | $62.762_{\pm 21.212}$ | $\mathbf{8.748}_{\pm 14.777}$ | $\mathbf{2.964}_{\pm 1.559}$ | $\mathbf{3.876}_{\pm 0.341}$ |
| | Sol | $\mathbf{70.112}_{\pm 16.912}$ | $\mathbf{3.242}_{\pm 2.362}$ | $\mathbf{0.512}_{\pm 1.556}$ | $3.803_{\pm 0.321}$ |

Specifically, we find that MemDLM performs strongly when infilling soluble motifs. Compared to EvoDiff, MemDLM-infilled soluble regions achieve higher pLDDT scores, suggesting that MemDLM sequences are more likely to fold into structurally plausible configurations. In addition, MemDLM attains lower pseudo-perplexity than both EvoDiff and the test set, indicating that the model recovers soluble residues with greater confidence. Finally, MemDLM-generated scaffolds also achieve higher BLOSUM-62 scores relative to EvoDiff, reflecting that the recovered sequences are more biologically conserved and closer to natural protein distributions.

### 3.4 TOKEN-LEVEL DISCRETE DIFFUSION GUIDANCE

**Setup** Solubilizing membrane proteins is an important therapeutic design task to improve the efficacy of drug delivery systems. Consequently, we apply our PET algorithm to optimize specific insoluble amino acid tokens during inference. We take the 1,098 experimentally-verified membrane protein sequences in the MemDLM test set and mask out the annotated TM residues. We input these partially masked sequences into MemDLM and use the PET algorithm to derive soluble analogs of the original membrane proteins while preserving the initial sequence scaffold.

**Results** Our results (Table 3, Supplementary Figure A4) demonstrate that sequences infilled with MemDLM and PET achieve a lower TM Residue Density compared to the original membrane protein while still preserving critical TM amino acid tokens.

Table 3: Computational validation of 1,098 membrane proteins solubilized from the test set using MemDLM with PET sampling. TMRD denotes the TM Residue Density Metric.

|  | PLDDT ($\uparrow$) | TMRD ($\downarrow$) | PPL ($\downarrow$) | BLOSUM ($\uparrow$) | ENTROPY ($\uparrow$) |
|---|---|---|---|---|---|
| Test Set | $76.637_{\pm 10.676}$ | $0.294_{\pm 0.219}$ | $5.707_{\pm 3.435}$ | $-$ | $3.918_{\pm 0.253}$ |
| MemDLM | $62.979_{\pm 17.906}$ | $0.181_{\pm 0.192}$ | $8.472_{\pm 4.879}$ | $0.495_{\pm 2.346}$ | $3.870_{\pm 0.268}$ |

At the same time, our sequences achieve a BLOSUM-62 score that closely matches the score obtained when MemDLM unconditionally infills the soluble domain (Table 2). This shows that applying PET with MemDLM not only solubilizes the protein but also favors biologically conserved amino acids, similar to what is observed during unconditional soluble motif scaffolding. Moreover, MemDLM-solubilized sequences maintain pseudo perplexity and entropy values similar to those of unconditionally generated samples, indicating that overall naturalness is preserved under the solubilization scheme. Supplementary Figure D further illustrates this with AlphaFold3 visualizations of original proteins and their solubilized counterparts, where a distinct loop extends from the main $\alpha$-helical core, likely corresponding to an extracellular soluble domain. Taken together, these results validate PET as an effective algorithm for token-level discrete diffusion guidance, successfully steering MemDLM to solubilize membrane proteins while preserving the likelihood of the original sequence.

## 4 DISCUSSION

In this work, we introduce **MemDLM**, the first classifier-guided discrete diffusion language model for *de novo* membrane protein design. By leveraging masked diffusion, MemDLM captures long-range dependencies essential to membrane protein structure and function – an area where structure-based models often fall short due to their reliance on fixed templates and limited generative flexibility. Our Per-Token Guidance (**PET**) framework enables targeted solubilization of membrane proteins while preserving key TM scaffolds. MemDLM also outperforms existing models in scaffolding functional motifs, maintaining biological relevance, and recovering native-like sequences. Looking ahead, we aim to extend MemDLM to generate diverse membrane topologies, including $\beta$-barrel and higher-order states (Qing et al., 2020; Mravic et al., 2024), and continue experimental validation of its designs. By evaluating both the structural fidelity and functional effects of PET-optimized sequences, we will further demonstrate MemDLM's utility in rational membrane protein engineering and therapeutic development.

REPRODUCIBILITY STATEMENT

We have taken extensive steps to ensure the reproducibility of our work. The MemDLM dataset was curated from publicly available membrane protein structural databases (PDBTM, OPM, mpstruc, and MemProtMD) following strict resolution and redundancy criteria, with final train/validation/test splits clearly documented (Supplementary Section A.1). All model architectures (EvoFlow backbone for MemDLM, solubility classifier, and baselines including DPLM, ProGen2, and EvoDiff) are described in detail, including hyperparameters, optimization schemes, and training schedules (Supplementary Section B.2). Computational benchmarks rely on standard metrics such as pLDDT, pseudo-perplexity, Shannon entropy, BLOSUM-62, and TM residue density (Tables 2 and 3) with code provided for metric calculation. Experimental protocols, including cloning, plasmid construction, and TOXCAT–$\beta$-lactamase growth assays, are described step by step with plasmid maps, sequences, and antibiotic selection conditions (Figure 2, Supplementary Figures A5–A7). We will release all code, data splits, pretrained model checkpoints, and experimental constructs upon publication to enable full reproducibility.

ETHICS STATEMENT

This work is focused on the development of generative models for membrane protein design, with potential applications in therapeutics, drug delivery, and synthetic biology. All computational experiments rely exclusively on publicly available protein sequence and structural databases, and no personally identifiable, clinical, or sensitive human subject data were used. Experimental validation was performed in *E. coli* using standard laboratory assays (TOXCAT–$\beta$-lactamase; Figure 2, Table 6), which do not raise ethical concerns regarding animal or human research. Potential risks of misuse, such as the creation of harmful proteins, are mitigated by the focus on therapeutic protein design and by releasing datasets and code under responsible-use licenses. We believe the societal benefits of improved protein design tools for medicine and biotechnology outweigh potential risks, and we encourage the community to adopt similar safeguards when extending this work.

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

# A EXTENDED BACKGROUND

## A.1 LANGUAGE MODELING

**Masked Language Models**   Masked Language Models (MLMs) employ Transformer-based architectures to learn bi-directional sequence context, distant token relationships, and predict the identity of corrupted (masked) amino acid tokens. The model is trained under a sequence-recovery training objective:

$$\mathcal{L}_{\text{MLM}} = -\sum_{i \in \mathcal{M}} \log p_\theta(x^i | x^{\setminus \mathcal{M}}) \tag{10}$$

where the set of masked positions $\mathcal{M}$ is a fraction of the sequence tokens. MLMs are strong representation-learners and excel at understanding both protein and natural languages. However, training these models to reconstruct only a minor fraction of tokens (15-40%) across a sequence makes complete *de novo* sequence generation difficult. (Devlin, 2018) (Lin et al., 2023) (Vincoff et al., 2025).

**Autoregression**   AR language models apply the chain rule to obtain a sequential factorization. These models are trained to maximize the log-likelihood of the data:

$$\mathbb{E}_{q(\mathbf{x})} \log p_\theta(\mathbf{x}) = \mathbb{E}_{q(\mathbf{x})} \sum_{i=1}^{L} \log p_\theta(\mathbf{x}^i | \mathbf{x}^{1:L}) \tag{11}$$

New samples can be drawn ancestrally in $L$ steps ($x^1 \sim p_\theta(x^1), \ldots, x^L \sim p_\theta(x^L | x^{1:L-1})$) ) following a strictly left-to-right unidirectional protocol. These models are a viable choice for natural language modeling schemes where a linear relationship between past and present values is inherently assumed. However, in biological contexts, such as protein sequences, AR models are limited by their inability to capture non-linear and long-range dependencies. For example, multi-pass membrane proteins consist of interleaved TM and soluble regions that are spatially and functionally coupled but may be separated by long sequence distances.

**Denoising Diffusion Models**   Diffusion models are a class of generative models defined by Markov processes (Ho et al., 2020) (Sohl-Dickstein et al., 2015). The *forward* diffusion steps $q(\mathbf{x}_{1:T} | \mathbf{x}_0) = \prod_{t=1}^{T} q(\mathbf{x}_t | \mathbf{x}_{t-1})$ progressively corrupt an initial data sample $\mathbf{x}_0 \sim q(\mathbf{x}_0)$ into a noisy prior $\mathbf{x}_T \sim q_{\text{noise}}$ across $T$ timesteps. The noise distribution $q_{\text{noise}}$ typically corresponds to an isotropic Gaussian, $\mathcal{N}(0, I)$, in continuous latent spaces, or a uniform categorical distribution over the vocabulary, $\text{Cat}(|\mathcal{V}|)$, in the discrete case. During inference, the learned *backward* process $p_\theta(\mathbf{x}_{0:T}) = p(\mathbf{x}_t) \prod_{t=1}^{T} p_\theta(\mathbf{x}_{t-1} | \mathbf{x}_t)$ gradually denoises the corrupted data sample to obtain samples from the true data distribution. Diffusion models are trained to maximize the evidence lower bound (ELBO):

$$\mathbb{E}_{q(\mathbf{x}_0)} \left[ \log p_\theta(\mathbf{x}_0) \right] \geq \mathbb{E}_{q(\mathbf{x}_{0:T})} \left[ \log \frac{p_\theta(\mathbf{x}_{0:T})}{q(\mathbf{x}_{1:T} \mid \mathbf{x}_0)} \right]$$

$$= \mathbb{E}_{q(\mathbf{x}_0)} \left[ \log p_\theta(\mathbf{x}_0 \mid \mathbf{x}_1) + \text{const.} - \sum_{t=2}^{T} \underbrace{\text{KL} \left( q(\mathbf{x}_{t-1} \mid \mathbf{x}_t, \mathbf{x}_0) \parallel p_\theta(\mathbf{x}_{t-1} \mid \mathbf{x}_t) \right)}_{\mathcal{F}_t} \right] \tag{12}$$

New data samples can be drawn by sampling from $q_{\text{noise}}(\mathbf{x}_T)$ and iteratively applying the learned denoising process $p_\theta(\mathbf{x}_{t-1}) = p_\theta(\mathbf{x}_{t-1} | \mathbf{x}_t)$. Various authors ((Sahoo et al., 2024), (Zheng et al., 2023)) have made simplifying assumptions about the reverse process to derive a computationally inexpensive loss function that reduces to a weighted negative log-likelihood, akin to a weighted form of Eq. 10.

## A.2 CLASSIFIER-GUIDED SAMPLING

**Preliminaries** Given a property $y$, guided diffusion aims to maximize $q(y|\mathbf{x})$ by sampling from the joint distribution $\mathbf{x} \sim q(\mathbf{x}_0, y)$. Therefore, the reverse transition can be conditioned on the property value $y$ and prior sequence samples. Using Bayes theorem, the conditional joint distribution can be decomposed:

$$q(\mathbf{x}_{t-1}|\mathbf{x}_t, y) = \frac{q(y|\mathbf{x}_{t-1}, \mathbf{x}_t)}{q(y|\mathbf{x}_t)} \tag{13}$$

In practice, the true distribution of $q(y|\mathbf{x}_t)$ is unknown and can be learned with a neural network $p_\phi(y|\mathbf{x}_t)$. To yield a tractable marginal reverse transition from Eq. 13, we can substitute the true distribution $q(\cdot)$ with our learned neural networks:

$$p_{\theta,\phi}(\mathbf{x}_{t-1}|\mathbf{x}_t, y) = \frac{p_\theta(y|\mathbf{x}_{t-1}, \mathbf{x}_t)}{p_\phi(y|\mathbf{x}_t)} \tag{14}$$

The normalization term in the denominator $p_\phi(y|\mathbf{x}_t)$ can be safely dropped since the model's parameters learn the normalized distribution. We can update the parameters $\theta, \phi$ at each iteration in the direction given by the gradient

$$\nabla_{\mathbf{x}_{t-1}} \log p_{\theta,\phi}(\mathbf{x}_{t-1}|\mathbf{x}_t, y) = \nabla_{\mathbf{x}_{t-1}} \log p_\phi(y|\mathbf{x}_{t-1}) + \nabla_{\mathbf{x}_{t-1}} \log p_\theta(\mathbf{x}_{t-1}|\mathbf{x}_t) \tag{15}$$

With this formulation, we can steer the denoising trajectory of the unconditional diffusion model to maximize the target attribute $y$ using gradients from an external classifier (Dhariwal & Nichol, 2021). Unlike classifier-free guidance, classifier-guidance prevents expensive retraining of existing denoising network on high-quality, task-specific labeled data and opens avenues for flexible, plug-and-play conditioning for various downstream applications.

**Discrete Classifier Guidance** While classifier guidance is well-formulated for diffusion models that operate over continuous data in Euclidean space (Dhariwal & Nichol, 2021), applying it to discrete spaces requires additional approximation. One common approach treats discrete tokens as continuous relaxations on the probability simplex and uses a first-order Taylor expansion around $\mathbf{x}_t$ to approximate $\log p_\phi(y|\mathbf{x}_{t-1})$ by making $\nabla_{\mathbf{x}_t}(\cdot)$ a valid operator. However, this approximation can be inaccurate when the local linearization poorly captures the classifier's behavior over discrete transitions, especially in regions with sharp decision boundaries. To remedy this, several methods ((Li et al., 2024), (Vignac et al., 2022)) have been proposed to circumvent the lack of continuous representations in discrete gradient guidance; most relevant to our work is LaMBO-2 introduced by (Gruver et al., 2024).

**LaMBO-2** To realize classifier-guidance for discrete sequences, LaMBO-2 first conducts sequence optimization using a Langevin process over a property-informed latent space. We begin with the discrete Langevin dynamics used in score-based models:

$$\mathbf{x}'_t = \mathbf{x}_t - \eta \nabla_{\mathbf{x}} \log p_\theta(y \mid \mathbf{x}_t) + \sqrt{2\eta\tau}\boldsymbol{\epsilon}, \quad \boldsymbol{\epsilon} \sim \mathcal{N}(0, I), \tag{16}$$

and generalize this update to the continuous latent space $h'_t \in \mathbb{R}^{1 \times D}$ guided by a differentiable surrogate of the discrete generative model. The batch size dimension $B$ is set to 1 for simplicity. The latent update step is defined as:

$$h'_t \leftarrow h'_t - \eta \nabla_{h'_t} [\lambda \mathrm{KL}(p_\theta(\mathbf{x}_t|h'_t) \,||\, p_\theta(\mathbf{x}_t|h_t)) - \sigma(v_\theta(h'_t)_d)] + \sqrt{2\eta\tau}\boldsymbol{\epsilon}, \quad \boldsymbol{\epsilon} \sim \mathcal{N}(0, I) \tag{17}$$

with step size $\eta$, temperature $\tau$, and regularization strength $\lambda$, where the sigmoid operator $\sigma(\cdot)$ can be applied to produce a sequence-level binary class probability from the classifier's unnormalized logit. The explore-exploit loss $\mathcal{L}_{\mathrm{EE}} := \lambda[\mathrm{KL}(p_\theta(\mathbf{x}_t|h'_t) \,||\, p_\theta(\mathbf{x}_t|h_t)) - \sigma(v_\theta(h'_t)_d)]$ guides the latent representation towards high values of the property with the gradient $\nabla_h \sigma(v_\theta(h))$, while the KL

term ensures the transition distribution maximizes the original sequence likelihood. Given a discrete sequence $\mathbf{x}_t$ and its corresponding latent representation $h_t$, one can take $N$ Langevin steps of Eq. 17 to realize optimized sequence latent representations before using the language-modeling head of the denoising network to project continuous embeddings to the discrete logit space ((Gruver et al., 2024), Appendix B.2). However, this construction does not guarantee the retention of specific tokens during inference because even if gradients are suppressed for particular positions, the subsequent projection through the language modeling head back into discrete logits does not ensure that the tokens with minimal gradient updates will be preserved.

## B  EXTENDED METHODS

### B.1  DATASET CURATION

**MemDLM**  Bioassembly structures from X-ray scattering or electron microscopy with better than 3.5 Å resolution, annotated by PDBTM1, mpstruc2, OPM3, or MemProtMD4, were used to curate membrane protein sequences for fine-tuning. *De novo* designed membrane proteins were added manually to the database. The proteins were culled at 100% sequence identity and 30% sequence identity to result in a non-redundant set and a sequence-diverse set, respectively. Integral membrane residues, defined as residues with at least one atom within the bilayer, were parsed from the resulting bioassembly structures using the membrane boundaries predicted by PPM 3.0 (Lomize et al., 2021). From the dataset of integral membrane residues, only structures with at least one TM chain spanning the entire membrane bilayer were included in the dataset. Additionally, chains without integral membrane residues were removed from the structure. All peripheral membrane proteins, defined as proteins with no TM chain, were filtered out. The TM protein sequences at the two sequence identity cut-offs and the Python script that parses the sequences from the PPM predictions are included in the SI. After these steps, 9,329 sequences with corresponding per-residue annotations remained. To augment this set of sequences, we obtained 2,579 unique PDB IDs from the Orientations of Proteins in Membranes (OPM) database with the provided "subunits" file (Lomize et al., 2006). PDB IDs were converted to corresponding protein sequences and per-residue labels (TM or soluble) were assigned using the subunits file. The final set of 11,908 TM sequences were then split using the MMSeqs2 easy clustering module with a minimum sequence identity of 80% and a coverage threshold of 50%. The resulting clusters were split to an 80-10-10 ratio into the training set (9,802 sequences, 82.31%), the validation set (1,008 sequences, 8.47%), and the testing set (1,098 sequences, 9.22%).

**PET Sampling Classifier**  We leveraged the same train/test/val set of 11,908 membrane sequences from the MemDLM dataset to develop a binary classifier that predicts the solubility of each amino acid within a protein sequence. Each sequence was annotated on a per-residue basis, with TM (class 1) and soluble (class 0) labels assigned according to the sequence's uppercase and lowercase residues, respectively.

### B.2  MODELING MEMDLM

**Model Architecture**  EvoFlow is a protein language model consisting of 33 Transformer-encoder layers and a language modeling head that is capable of *de novo* generating protein sequences. More formally, it can denoise a protein sequence consisting of all [MASK] tokens, making it a natural choice for a discrete diffusion-based protein language model. We use the pre-trained EvoFlow protein language model checkpoint (`https://huggingface.co/fredzzp/EvoFlow-650M-context-3070`) as the basis of our neural network $p_\theta$ since EvoFlow was trained under the RDM framework (forward process as defined by Eq. 1 and loss computation defined by Eq. 3). The Diffusion Protein Language Model (DPLM) was also trained under the RDM framework by (Wang et al., 2024) and is thereby an alternative choice for $p_\theta$. However, we opt for EvoFlow over DPLM as the architecture for $p_\theta$ as DPLM is restricted by its shorter context length of 1,024 tokens, compared to EvoFlow's extended context length of 3,070 tokens.

**Training**  To achieve membrane protein-specific generation, we fine-tuned EvoFlow by selectively updating a subset of the encoder's attention layers. Specifically, the final $N = 3$ Transformer encoder layers $\{\mathcal{L}_{M-N+1}, \ldots, \mathcal{L}_M\}$ are partially unfrozen, where $M = 33$ is the total number of encoder layers. Within each layer, we enable gradient updates to only the key, query, and value projection

matrices ($W_K$, $W_Q$, and $W_V$) of the self-attention mechanism and keep all other weights frozen. With this training recipe, we bias the pre-existing EvoFlow latent space with physicochemical features of membrane proteins without overfitting on the new sequences. MemDLM was trained to minimize the objective in Eq. 3 on a 4xA6000 NVIDIA DGX server with 200 GB of shared VRAM for 3K steps using the AdamW optimizer (betas=($\beta_1 = 0.99, \beta_2 = 0.98$), weight decay $\lambda = 0.01$), a learning rate (LR) of $4 \times 10^{-5}$ with a cosine schedule (150 linearly-scheduled warmup steps, LR minimum = $1 \times 10^{-5}$).

### B.3 Per-Token Solubility Classifier

Let $v_\phi : \mathbb{R}^{B \times L \times D} \to \mathbb{R}^{B \times L}$ be a neural network trained to predict per-token solubility scores from continuous latent representations $h_t$. The model is trained using clean protein sequences $\mathbf{x}$ with corresponding binary per-residue solubility labels $\mathbf{y} \in \{0,1\}^L$ (0 = insoluble, 1 = soluble). Each input sequence is first embedded using the pretrained ESM-2-650M protein language model checkpoint (https://huggingface.co/facebook/esm2_t33_650M_UR50D) (Lin et al., 2023). The resulting contextualized token embeddings are passed through a lightweight classifier $v_\phi$ with the following architecture: (i) trainable 2-layer Transformer encoder Transformer$_\phi$; (ii) LayerNorm and dropout ($p = 0.5$); and (iii) a trainable 2-layer projection head MLP$_\phi$ outputs a scalar logit for each token position. All parameters in ESM-2 are frozen, and only the transformer encoder and MLP layers are updated during training. The classifier is optimized using a per-token binary cross-entropy loss with logits:

$$\mathcal{L}_{\text{BCE}}(\phi) = - \left[ y \cdot \log \sigma(z) + (1 - y) \cdot \log(1 - \sigma(z)) \right] \tag{18}$$

where $\sigma(z)$ is the sigmoid activation function and $\mathbf{z} = v_\phi(h)$ is a vector of per-token logit predictions. The loss is computed without reduction to allow for masking padded positions and is averaged over all valid tokens in the batch. $v_\phi$ is trained on a 1xA6000 NVIDIA DGX server with 50 GB of shared VRAM for 50K steps using the AdamW optimizer (betas=($\beta_1 = 0.99, \beta_2 = 0.98$), weight decay $\lambda = 0.01$), a learning rate (LR) of $3e^{-5}$ with a cosine schedule (5000 warmup steps, LR minimum = $1e^{-5}$). The PET classifier was trained using the same train, test, and validation sequence splits as MemDLM pre-training.

### B.4 Benchmarking Models

We fine-tune ProGen2 (Nijkamp et al., 2023) and Diffusion Protein Language Model (DPLM) (Wang et al., 2024) on the same train, test, and validation split of membrane protein sequences used to train MemDLM. We use these SOTA models along with EvoDiff as the basis for comparing MemDLM's performance on membrane protein design tasks.

#### B.4.1 ProGen2

The ProGen2 protein language consists of 27 Transformer layers and was trained under the autoregressive formulation (Supplementary Section A.1) to *de novo* generate protein sequences (Nijkamp et al., 2023). We fine-tune the pre-trained ProGen2-base 764M model checkpoint (https://huggingface.co/hugohrban/progen2-base) to achieve membrane protein-specific generation. Specifically, we unfreeze the final $N = 2$ Transformer layers and enable updates to only self-attention module and fine-tune the model for 5,000 steps. We use 2xH100 NVIDIA GPUs with 192 GB of shared VRAM, the Adam optimizer (betas=($\beta_1 = 0.9, \beta_2 = 0.999$), $\lambda = 0.1$, moving averages for the first and second moment estimators set to zero), and a learning rate of $2e^{-4}$ set to decay by a factor of 5, as detailed in Section 3.3 of (Nijkamp et al., 2023).

#### B.4.2 DPLM

DPLM is an RDM-based discrete diffusion protein language model consisting of 33 Transformer layers that can *de novo* generate protein sequences, scaffold over functional motifs, and produce protein sequence embeddings (Wang et al., 2024). We fine-tune the pre-trained DPLM 650M model checkpoint (https://huggingface.co/airkingbd/dplm_650m) on the RDM training objective (Eq. 3) to achieve membrane protein-specific generation. Specifically, we use 2xH100 NVIDIA GPUs with 192 GB of shared VRAM for 5K steps using the AdamW optimizer (betas=($\beta_1 = 0.9, \beta_2 = 0.98$), weight decay $\lambda = 0.01$), a learning rate (LR) of $4 \times 10^{-5}$ with a cosine schedule

(150 linearly-scheduled warmup steps, LR minimum = $1 \times 10^{-5}$) as detailed in the provided DPLM source code (https://github.com/bytedance/dplm).

### B.4.3 EVODIFF

We elect to use EvoDiff under the Order-Agnostic Autoregressive Diffusion sampling framework (EvoDiff-OADM) over EvoDiff trained under the Discrete Denoising Diffusion Probabilistic Model (Austin et al., 2021) (EvoDiff-D3PM) as EvoDiff-OADM was trained to denoise sequences consisting of masked tokens and EvoDiff-D3PM was trained to denoise starting from a sequence of uniform amino acids. We use the provided pre-trained EvoFlow-OADM-640M checkpoint and sampling code (https://github.com/microsoft/evodiff) for EvoDiff benchmarking.

### B.5 COMPUTATIONAL METRICS

Sequence generation quality was computationally verified using the following metrics:

**Pseudo Perplexity**   The model's generation quality was assessed using the ESM-2 (Lin et al., 2023) pseudo-perplexity metric. Typically, a lower pseudo-perplexity value indicates higher confidence. Specifically, the pseudo-perplexity is computed as the exponential of the negative pseudo-loglikelihood of a sequence. This metric yields a deterministic value for each sequence but necessitates L forward passes for computation, where L represents the input sequence length. It is formally defined as $\text{PPL}(\mathbf{x}) = \exp[-\frac{1}{L} \sum_{i=1}^{L} \log p(x^i \mid x^{\backslash i})]$.

**pLDDT**   The structural confidence of generated sequences was assessed using predicted Local Distance Difference Test (pLDDT) scores from ESMFold v1 with chunk size of 128 (Lin et al., 2023), a protein language model-based tool to predict protein structures from amino acid sequences alone. Higher pLDDT indicates ESMFold is more confident in the produced structure, suggesting the initial input sequence is biologically plausible.

**Shannon Entropy**   To measure the diversity and uncertainty of the model's token predictions, we compute the average Shannon entropy across the sequence. Let $p(x^i)$ denote the model's probability distribution over the vocabulary $\mathcal{V}$ at position $i$. Higher entropy values indicate greater diversity in the model's predictions, while lower values suggest more repetitive distributions. The entropy is defined as: $\text{Entropy}(\mathbf{x}) = -\frac{1}{L} \sum_{i=1}^{L} \sum_{v \in \mathcal{V}} p(x^i = v) \cdot \log p(x^i = v)$.

**BLOSUM62 Substituion Score**   The average BLOSUM62 score is a quantitative approach to determining whether an amino acid substitution is conservative or nonconservative. This value becomes an important computational metric for protein sequence infilling tasks (both unconditional and PET-based solubilization) to determine if the model is introducing non-conserved residue changes. For each aligned position between a generated sequence $\hat{\mathbf{x}}$ and reference sequence $\mathbf{x}$, we extract the substitution score $B(\hat{x}^i, x^i)$ from the BLOSUM62 matrix (Henikoff & Henikoff, 1992). Higher scores indicate greater biochemical similarity to the native sequence, while lower scores suggest more divergent or potentially deleterious substitutions. The final score is computed as the mean over all aligned residues $\text{BLOSUM}(\hat{\mathbf{x}}, \mathbf{x}) = \frac{1}{L} \sum_{i=1}^{L} B(\hat{x}^i, x^i)$.

**TM Residue Density**   To estimate the membrane-localizing potential of generated sequences, we used DeepTMHMM v1.0 tool (https://services.healthtech.dtu.dk/services/DeepTMHMM-1.0/) (Hallgren et al., 2022) to produce per-residue topology annotations. Each residue is classified into one of six categories: signal peptide (S), inside cell/cytosol (I), alpha membrane (M), beta membrane (B), periplasm (P), or outside cell/lumen (O). For our analysis, we consider residues labeled as alpha membrane (M) to be "soluble" in the membrane context, and all other classes, including beta membrane (B), to be "insoluble." We explicitly exclude B-labeled residues from the soluble category due to the structural and biophysical differences between beta-barrel and alpha-helical transmembrane domains, the latter being dominant in our training set. Using these annotations, we define the *TM Residue Density* of a sequence as the number of residues predicted to lie within alpha membrane ("M" predictions) regions divided by the sequence length as a normalization factor.

## B.6 WET-LAB EXPERIMENTS

### B.6.1 CLONING AND PLASMID CONSTRUCTION

DNA sequences of our MemDLM-designed and control peptides were cloned into the pMAL_dst$\beta$L vector (Addgene plasmid #73805) between the genes encoding for ToxR and $\beta$-lactamase using blunt-end ligation. The resulting constructs were initially transformed into *E. coli* XL-10 Gold cells. Transformants were selected on Luria Broth (LB) agar plates containing spectinomycin and sequences were verified by Sanger sequencing. Confirmed plasmids were subsequently transformed into *E. coli* Cloni cells for the assay.

Cell lines:

| REAGENT | CATALOG INFORMATION |
|---|---|
| E. Cloni 10G DUOs Chemically Competent Cells | Cat. No. 60107-1 (BioSearch Technologies) |
| XL 10-Gold Ultracompetent Cells | Cat. No. 200315 (Agilent) |

Table 4: Competent cell reagents used in this study.

Genes inserted into the pMAL_dst$\beta$L plasmid vector:

- **Human CLS:**
  - UniProt: UPI000007083D
  - Amino acid sequence: PLFIPVAVMVTAFSGLAFIIWLA
  - Gene: CCGCTGTTCATCCCGGTTGCAGTTATGGTTACCGCTTTTAGTGGATTG-GCGTTTATCATCTGGCTGGCT
- **GpA-TM Region:**
  - UniProt: UPI000012B75E
  - Amino acid sequence: LIIFGVMAGVIGTILI
  - Gene: TTAATTATTTTCGGAGTGATGGCCGGAGTTATCGGCACAATTTTAATC
- **ErbB2 TM Region:**
  - UniProt: P04626-1
  - Amino acid sequence: SIISAVVGILLVVVLGVVFGIL
  - Gene: TCCATTATCTCCGCTGTCGTAGGAATCTTGTTAGTTGTCGTC-CTTGGGGTTGTGTTTGGAATTTTA
- **Qsox2 TM Region:**
  - UniProt: Q6ZRP7
  - Amino acid sequence: SLCVVLYVASSLFMVMYFF
  - Gene: AGTCTTTGCGTCGTACTTTACGTCGCATCTTCACTGTTTATGGTGATG-TATTTCTTT
- **EK3 Water Soluble Helix** (Wolny et al., 2017):
  - Amino acid sequence: SAEEEKKKAEEEKKKAEEEKKKAE
  - Gene: TCCGCAGAGGAAGAAAAGAAAAAGCTGAAGAAGAAAAGAAAAAG-GCAGAAGAAGAGAAAAAAAAGGCAGAG
- **PoorTM2**
  - MemDLM amino acid sequence: SSLLFSYQGAKKEEERVFLDNF
  - Gene: AGTTCTTTGTTATTCAGCTATCAGGGAGCCAAGAAAGAAGAA-GAACGTGTGTTTCTGGATAACTTC
- **PoorTM4**
  - MemDLM amino acid sequence: GTHAKDWRVTSWKRYGEIE
  - Gene: GGAACACATGCTAAAGATTGGCGTGTGACATCTTGGAAGCGTTACG-GCGAGATTGAA

- **GoodTM4**

  – MemDLM amino acid sequence: DLSKWLGIVLLLLLAILALLLIR

  – Gene: GATTTAAGCAAATGGCTGGGTATCGTACTGTTACTGTTACTGGC-
    TATTTTGGCTTTATTACTGATTCGT

- **GoodTM5**

  – MemDLM amino acid sequence: SLRWLWSLVIGLLLIVAFYLLLR

  – Gene: AGCCTGCGTTGGTTGTGGTCTTTAGTGATCGGCTTACTGCT-
    TATCGTTGCCTTCTACCTGCTGCTTCGC

- **GoodTM8**

  – MemDLM amino acid sequence: DFLRKAVIVLLVLVIVAGLLVIR

  – Gene: GATTTTCTGCGTAAGGCAGTGATTGTATTACTTGTCTTGGTTATTGTG-
    GCGGGTCTGCTGGTTATTCGC

### B.6.2 TOXCAT-$\beta$-Lactamase Growth Assay

Single colonies of plasmid-containing E. *coli* Cloni cells were used to inoculate 6-mL LB cultures supplemented with 50 $\mu$g/mL spectinomycin. Glycerol stocks were made and used to inoculate new fresh LB culture tubes with 50 $\mu$g/mL spectinomycin. Cultures were incubated for ∼8 h or overnight at 37°C with shaking. Optical density at 600 nm (OD$_{600}$) was measured, and cultures were diluted with fresh LB + spectinomycin to an OD$_{600}$ of 0.05. Growth was continued until an OD$_{600}$ of ∼0.1 was reached.

To ensure consistent inoculation density across assays, the number of cells per well was normalized to $1.95 \times 10^5$ cells. This value was calculated using the relationship of 1 OD$_{600}$ ≈ $8 \times 10^8$ cells/mL and adjusted for the measured absorbance at OD$_{600}$ of each culture. Growth under spectinomycin confirmed that the pMal_dsTBL plasmid was successfully introduced into E. *coli* Cloni cells across all conditions. All cultures grew equally under this condition, demonstrating comparable inoculation densities and consistent plasmid uptake.

Assays were performed in 96-well plates, with each well containing a final total volume of ∼200 $\mu$L LB medium supplemented with the appropriate antibiotics in the following concentrations: Spectinomycin (50 $\mu$g/mL), Carbenicillin (300 $\mu$g/mL), Carbenicillin (100 $\mu$g/mL) + chloramphenicol (100 $\mu$g/mL), Carbenicillin (100 $\mu$g/mL) + chloramphenicol (120 $\mu$g/mL). Wells were inoculated with the calculated volume of diluted culture corresponding to $1.95 \times 10^5$ cells. Each antibiotic reporter was run in triplicate. Plates were incubated at 37°C in a pre-heated plate reader (BioTek Synergy H1). Bacterial growth was monitored by measuring absorbance at 600 nm for 24 hours with measurements taken every 10 minutes under continuous shaking.

## C EXTENDED RESULTS

### C.1 DENSITY PLOTS

We visualize the density distribution of the various computational metrics to assess membrane protein sequences. When using P2 Self-Planning to generate sequences, we set $\tau = 0.7$ to have a slight bias towards deterministic model outputs.

**Unconditional Generation** We visualize the density distribution of unconditionally generated membrane proteins from various models.

Figure A1: *De novo*-generated and natural membrane protein sequences from various models. **A)** TM Residue Density predicted by DeepTMHMM. **B)** ESM-2-650M Pseudo Perplexity. **C)** ESMFold-predicted pLDDT scores. **D)** Shannon entropy values.

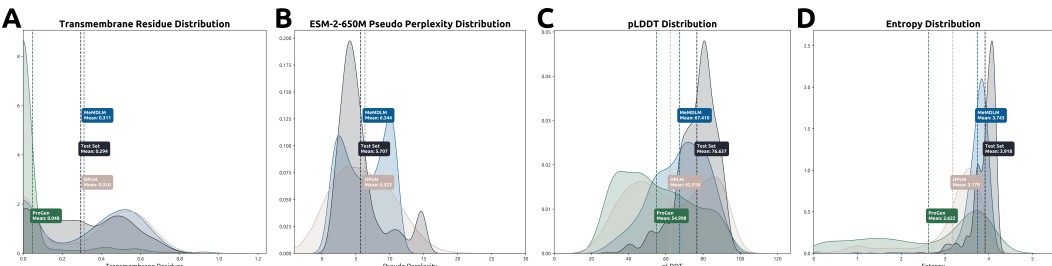

**Motif Scaffolding**    We mask out and infill both the insoluble and soluble regions of natural membrane proteins derived from the model's test set.

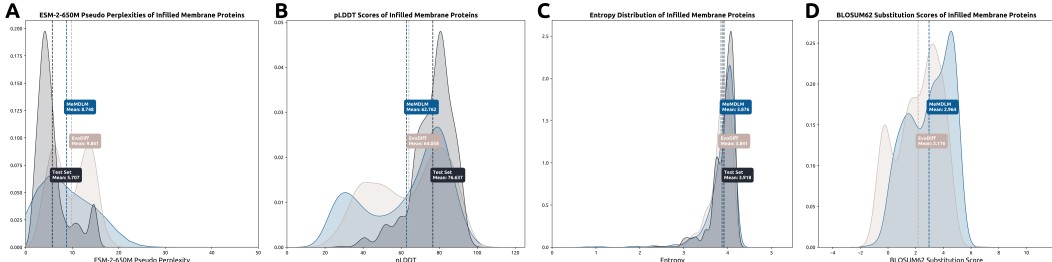

Figure A2: Infilling Insoluble Domain

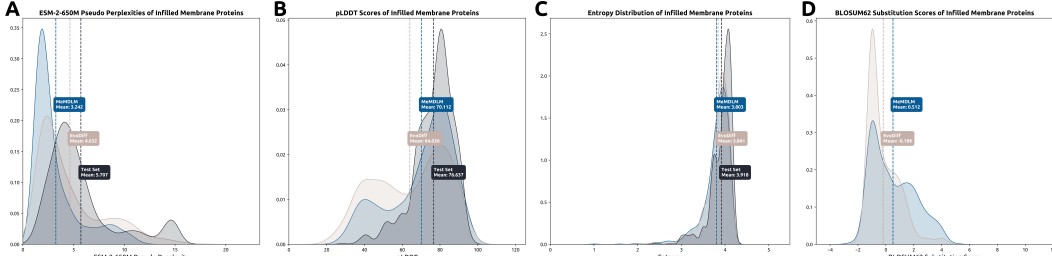

Figure A3: Infilling Soluble Domain

**Solubilization**    We optimize the solubility of the proteins in the model's test set by applying our PET algorithm.

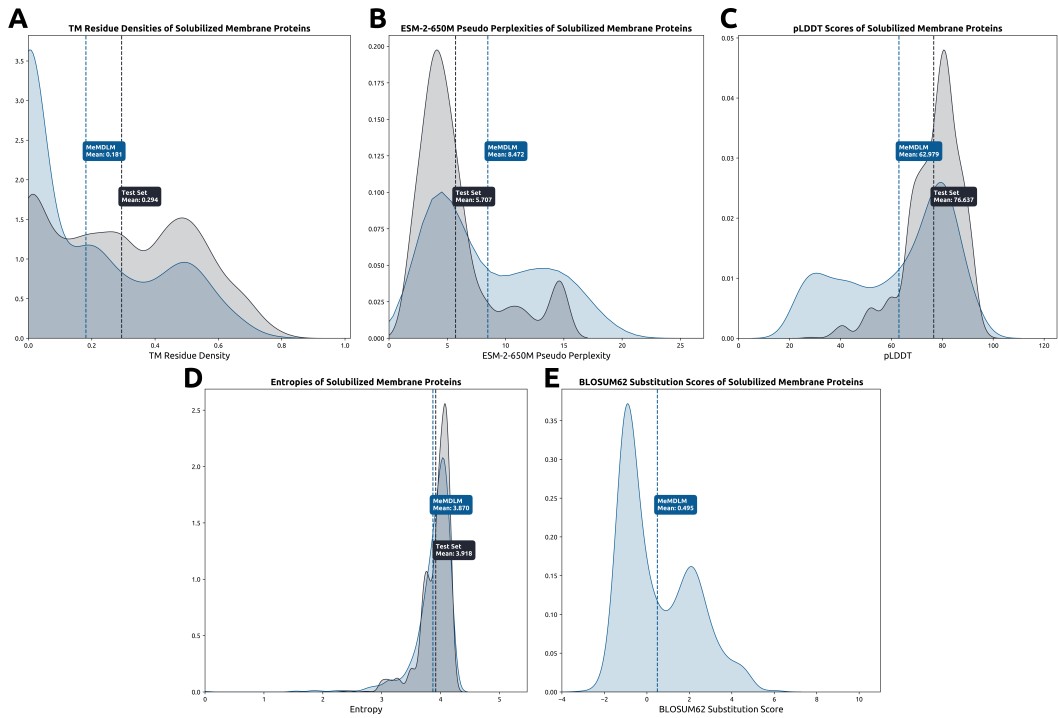

Figure A4: Solubilizing TM Domains

## C.2 Physiochemical Property Prediction

As a surrogate task, we assessed if RDM training retains physicochemical information critical to membrane protein function by predicting per-residue solubility and membrane localization (Table 3). We use embeddings from three models–vanilla ESM-2-650M, ESM-2-650M fine-tuned on membrane protein sequences, and MemDLM–as inputs to a per-residue solubility and sequence-level membrane localization classifiers. We outline the dataset, training details, and evaluation results of these models in the following.

### C.2.1 Datasests

**Solubility Prediction** We leveraged the same set of 11,908 membrane sequences from the MemDLM training dataset to develop a binary classifier that predicts the solubility of each amino acid within a protein sequence. Each sequence was annotated on a per-residue basis, with TM (class 1) and soluble (class 0) labels assigned according to the sequence's uppercase and lowercase residues, respectively. The same training, testing, and validation data splits used to train MemDLM were also utilized to train and evaluate this classifier.

**Membrane Localization** We collected 30,020 protein sequences from DeepLoc 2.0 (Thumuluri et al., 2022) to build a binary classifier that predicts a protein sequence's cellular localization. The authors of the dataset provided a multi-label label for each sequence indicating its localization(s). We used the authors' provided data splits, with training sequences having 11 labels and testing sequences having 8 labels.

### C.2.2 Models

**Solubility Prediction** We first predicted TM and soluble residues, a hallmark characteristic of membrane protein sequences. We utilized embeddings from each pLM's latent space (ESM-2-150M, ESM-MLM, and MemDLM) as inputs to train a two-layer perceptron classifier that minimized the standard binary cross-entropy (BCE) loss to compute the probability that each residue in the sequence is either soluble (probability < 0.5, class 0) or TM (probability > 0.5, class 1).

**Membrane Localization Prediction**    Proteins originating from the endomembrane system and localizing in the plasma membrane differ in conformation and function from those in the cytosol and other cellular organelles. We predicted the subcellular localization of protein sequences by utilizing embeddings from each pLM's latent space (ESM-2-150M, ESM-MLM, and MemDLM) to train a XGBoost classifier that minimized the standard BCE loss to compute the probability that a protein sequence localizes in the plasma membrane (probability > 0.5, class 1) or in other regions (probability < 0.5, class 0).

**Fine-Tuning ESM-2**    We fine-tune the ESM-2 pLM ((Lin et al., 2023)) to achieve an encoder that produces membrane-aware protein sequence embedding used as a baseline comparison for the RDM training task. We trained a MLM head on top of ESM-2-650M using membrane protein sequences to force comprehension of membrane protein properties. We chose to randomly mask 40% of amino acid tokens during training over the standard 15% to more closely resemble the dynamics of diffusion-based (RDM) training; masking rates above 40% have been seen as detrimental during MLM training tasks (Wettig et al.). Corrupted sequences were passed into ESM-2-650M to retrieve their output embeddings. During training, we unfroze the key, query, and value weights in the attention heads of the final three encoder layers, similar to fine-tuning EvoFlow during MemDLM training. During ESM-2 fine-tuning, the model performed a *masked-prediction* task over masked amino acid tokens to minimize the NLL loss in Eq. equation 10. 2xH100 NVIDIA GPUs, learning rate of 5e-3, the Adam optimizer, and a batch size of 8 over 10 epochs were used.

### C.2.3    RESULTS

We leveraged the trained solubility prediction and membrane localization classifiers to determine if latent spaces from RDM-based generative models are aligned with relevant membrane protein properties. Table 5 shows that MemDLM latent embeddings achieve predictive performance that closely parallels SOTA pLM embeddings, which are designed specifically for delivering precise representations.

| MODEL | SOLUBILITY (↑) | MEMBRANE LOCALIZATION (↑) |
|---|---|---|
| ESM-2-650M | 0.9383 | 0.6011 |
| Fine-Tuned ESM-2 | 0.9375 | 0.6000 |
| MemDLM | 0.9375 | 0.5964 |

Table 5: Performance comparison (AUROC) of embeddings derived from various models in predicting physico-chemical properties of MemDLM test set sequences.

In total, these results demonstrate that MemDLM accurately captures the biological features underpinning functional membrane proteins despite being trained on a sequence generation task rather than a masked-prediction task.

## C.3 WET-LAB EXPERIMENTS

### C.3.1 TOXCAT ASSAY

| | ΔToxR-POI-βL | ΔPOI | Soluble | TM (monomer) | TM (dimer) |
|---|---|---|---|---|---|
| **Control Samples** | ΔToxR-POI-βL | ΔPOI | EK3 | CLS | ErbB2, GpA, Qsox2 |
| **Spectinomycin** | Growth | Growth | Growth | Growth | Growth |
| **Carbenicillin** | No growth | No growth | No growth | Growth | Growth |
| **Carbenicillin + Chloramphenicol** | No growth | No growth | No growth | No growth | Growth |

Figure A5: Summary of control constructs for the TOXCAT-$\beta$-lactamase assay and their expected growth responses to antibiotics.

Schematic showing gene ToxR-POI-$\beta$L, where POI is the peptide of interest and $\beta$L is $\beta$-lactamase. Periplasmic $\beta$-lactamase and cytoplasmic ToxR proteins are represented by blue and yellow dots, respectively. Expected growth phenotypes under spectinomycin and carbenicillin +/-chloramphenicol are indicated for each control. Negative controls ΔToxR-POI-$\beta$L, ΔPOI, and EK3 should not survive in carbenicillin because they lack a TM domain. Positive controls CLS, ErbB2, GpA, and Qsox2 all have TM domains and should survive in carbenicillin. Further, ErbB2, GpA, and Qsox2 are dimers. Expression of these controls should also confer resistance to chloramphenicol.

### C.3.2 TOXCAT SEQUENCE SELECTION

From 1,000 MemDLM-generated sequences, three sequences from the top 100 predicted performers ("GoodTM") and two sequences from the bottom 22 predicted performers ("PoorTM") were selected for screening in the TOXCAT assay. The following selection criteria was used:

| CATEGORY | PLDDT | PPL | TM RESIDUE DENSITY | SEQUENCES SELECTED |
|---|---|---|---|---|
| GoodTM (Top 100) | $> 60$ | $< 10$ | Non-zero | 3 |
| PoorTM (Bottom 22) | $< 60$ | $< 15$ | Non-zero | 2 |

Table 6: Selection criteria and sequence counts for MemDLM-generated sequences screened in the TOXCAT assay.

The top-ranked (GoodTM) sequences represent a diverse set of high-scoring designs. GoodTM4 (Sequence DLSKWLGIVLLLLLAILALLLIR, rank 41) contains high transmembrane residue density, GoodTM5 (Sequence SLRWLWSLVIGLLLLIVAFYLLLR, rank 57) contains a Small-X$_3$-Small motif known to promote TM-TM association (Russ & Engelman, 1999; Li et al., 2004; Russ & Engelman, 2000), and GoodTM8 (Sequence DFLRKAVIVLLVLVIVAGLLVIR, rank 8) has an increased abundance of charged residues capping the TM spanning domain. This further demonstrates that MemDLM generates plausible protein sequences with TM-like character.

### C.3.3 GROWTH CURVES

**Control Plasmids** Growth curves of *E. coli* Cloni cells containing control plasmids.

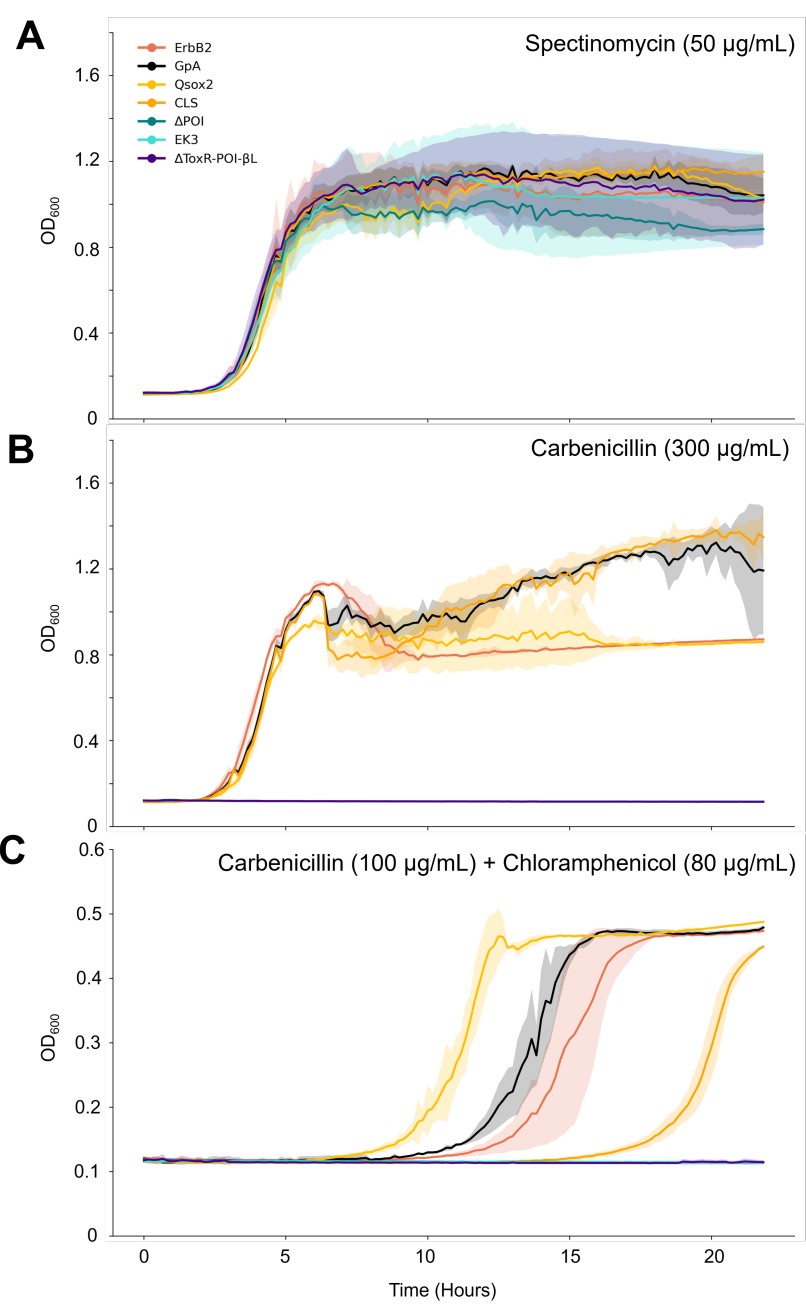

Figure A6: **A)** Survival in spectinomycin (50 $\mu$g/mL) confirmed plasmid uptake for all controls. **B)** Growth curves of control plasmids under carbenicillin (300 $\mu$g/mL) showed that control plasmids containing TM sequences survived selective pressure. **C)** Growth curves of control plasmids under combined carbenicillin (100 $\mu$g/mL) and chloramphenicol (80 $\mu$g/mL) selection, which tests both transmembrane insertion and association, show that the dimeric Qsox2, GpA, and ErbB2 controls begin growing in chloramphenicol earlier than the monomeric CLS control.

**MemDLM-Generated Sequences** Growth curves for MemDLM's *de novo*-generated sequences.

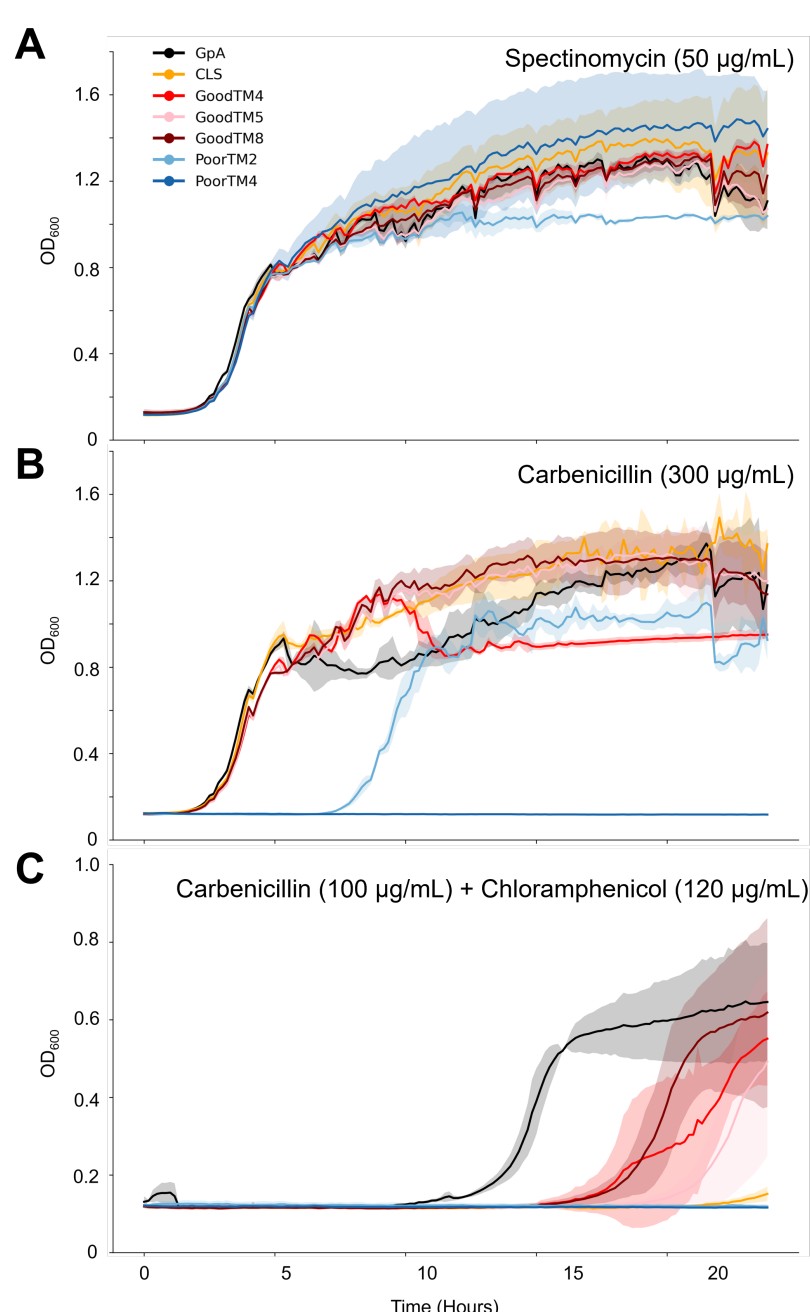

Figure A7: GpA is used as a positive control for insertion and TM association. CLS is the positive insertion and negative TM association control. **A)** Growth curve of *E. coli* Cloni cells containing *de novo* MemDLM TM sequences under spectinomycin (50 $\mu$g/mL) confirmed plasmid uptake. **B)** Growth curves of MemDLM peptides under carbenicillin (300 $\mu$g/mL) show GoodTM4, GoodTM5, and GoodTM8 growing as expected. PoorTM4 did not survive, indicating that it is not membrane inserting. PoorTM2 showed delayed growth, suggesting that it has lower membrane insertion propensity than the GoodTM constructs. **C)** Growth curves of MemDLM plasmids under combined carbenicillin (100 $\mu$g/mL) and chloramphenicol (120 $\mu$g/mL), used to select for both transmembrane insertion and transmembrane association, reveal that some of the TM designs may be oligomeric.

# D VISUALIZATIONS

AlphaFold3 visualizations of MemDLM-generated membrane protein sequences. TM Residue Density (TMRD) scores are derived from DeepTMHMM predictions. Structures and colors are from AlphaFold3 predictions, and pLDDT scores are from ESMFold.

## D.1 *De novo* GENERATION

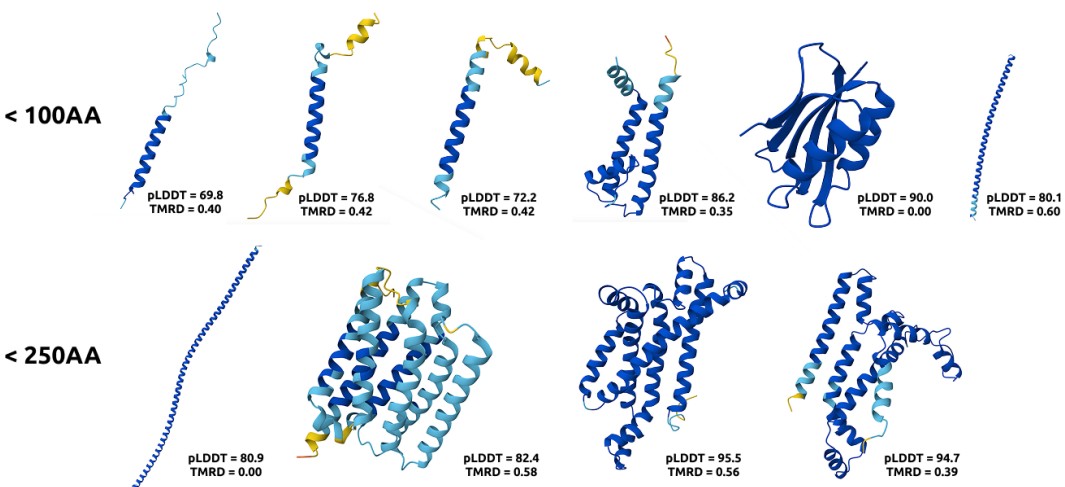

Figure A1: *De novo*-generated protein sequences from MemDLM across different lengths.

## D.2 SOLUBILIZATION

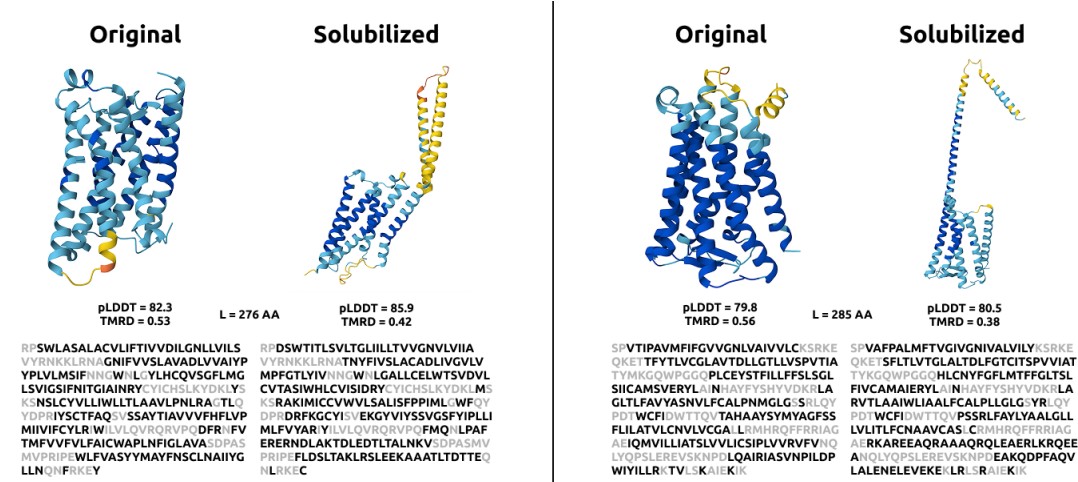

Figure A2: Original and solubilized versions of MemDLM test set protein sequences. Grey residues were annotated as soluble in the given sequence and were thus "fixed" during PET sampling.

## E  ALGORITHM PSEUDOCODE

---

**Algorithm 1** MemDLM Training

---

**Require:** Protein sequence dataset $\mathcal{D}$, diffusion model $p_\theta$, number of diffusion timesteps $T$

1: **while** not converged **do**
2:      Sample batch $\mathbf{x}_0 \sim \mathcal{D}$
3:      Sample timestep $t \sim \mathcal{U}(1, T)$
4:      Corrupt sequence: $\mathbf{x}_t \sim q(\mathbf{x}_t \mid \mathbf{x}_{t-1})$
5:      Compute RDM loss: $\mathcal{L}_{\text{RDM}} = -\lambda_t \sum_{i=1}^{L} \log p_\theta(x_0^i \mid \mathbf{x}_t)$
6:      Take gradient descent step on: $\nabla_\theta \mathcal{L}_{\text{RDM}}$
7: **end while**
8: **return** Trained MemDLM $p_\theta$

---

**Algorithm 2** MemDLM Sampling with P2 Self-Planning and Optional Sequence Refinement

---

**Require:** Fully masked sequence $\mathbf{x}_T = \{[\text{MASK}]\}_{i=1}^{L}$, trained MemDLM $p_\theta$, number of denoising steps $T$

1: **for** $t \in \{T, T-1, \ldots, 0\}$ **do**
2:      Compute logits: $\mathbf{z}_{t-1} = p_\theta(\mathbf{x}_t)$
3:      Sample candidate tokens: $x_{t-1}^i = \arg\max_v \left( \frac{z_{t-1}^{i,v}}{\tau} + g^{i,v} \right), \quad g^{i,v} \sim \text{Gumbel}(0,1)$
4:      Compute per-token log-probabilities: $s_t^i = \log p_\theta(x_t^i)$
5:      Identify unmasked positions: $\mathcal{R}_t = \{i \mid x_{t-1} \neq [\text{MASK}]\}$
6:      Compute $K = \lfloor (1 - \kappa_t) \cdot |\mathcal{R}_t| \rfloor$
7:      Select top-$K$ lowest scoring tokens from $\mathcal{R}_t$ and remask them: $x_t^i = [\text{MASK}]$ for $i \in$ top-$K(s_t^i)$
8:      Copy high-confidence predictions: $x_{t-1}^i \leftarrow x_t^i$ for positions previously masked but not in top-$K$
9: **end for**
10: **if** PET Optimization **then**
11:      Perform Algorithm 3
12: **end if**
13: **return** Final decoded sequence $\mathbf{x}_0$

---

**Algorithm 3** PET-based MemDLM Sampling

---

**Require:** Candidate protein sequence $\mathbf{x}$, trained MemDLM $p_\theta$, trained solubility classifier $v_\phi$, pre-trained encoder Encoder$_\phi$, number of optimization steps $N$

1: Produce sequence embeddings $h = \text{Encoder}_\phi(\mathbf{x})$
2: Compute saliency map $\mathbf{s}$ using gradients $\nabla_h v_\phi(h)$
3: Normalize saliency map $\hat{s}^i \leftarrow s_i$
4: Determine editable positions $\mathcal{E}$ based on soluble residues and saliency scores
5: **for** each $i \in \mathcal{E}$ **do**
6:      Define neighborhood $\mathcal{N}(i)$
7:      Compute $\tilde{s}^i = \hat{s}^i + \gamma \sum_{j \in \mathcal{N}(i)} \text{Norm}(A_{ij}) \cdot \hat{s}^j$
8:      Construct prior distribution $\pi(x^i)$
9:      Compute guidance distribution: $\log P(x^i) = (1 - \sigma(\alpha \tilde{s}^i)) \cdot \log p_\theta(x^i) + \sigma(\alpha \tilde{s}^i) \cdot \pi(x^i)$
10:      Sample token $\hat{x}^i \sim \text{CAT}(\log P(x^i))$
11:      Update $\mathbf{x}[i] \leftarrow \hat{x}^i$
12: **end for**
13: **return** Optimized sequence $\hat{\mathbf{x}}$

---

# F  USE OF LARGE LANGUAGE MODELS (LLMs)

We acknowledge the use of large language models (LLMs) to assist in polishing and editing parts of this manuscript. LLMs were used to refine phrasing, improve clarity, and ensure consistency of style across sections. All technical content, experiments, analyses, and conclusions were developed by the authors, with LLM support limited to language refinement and editorial improvements.

