# OpenReview forum: "Token-Level Guided Discrete Diffusion for Membrane Protein Design"
_ICLR.cc/2026/Conference — ICLR 2026 Conference Withdrawn Submission_

### Official Review · Reviewer_MoC8 · 2025-10-27

**Soundness:** 3
**Presentation:** 3
**Contribution:** 2
**Rating:** 6
**Confidence:** 3

**Summary:**

The manuscript proposes the Membrane Diffusion Language Model (MemDLM) for controllable membrane protein sequence design. A novel Per-Token Guidance strategy is introduced, enabling classifier-guided sampling that selectively solubilizes residues while preserving conserved transmembrane (TM) domains. This approach yields sequences with reduced TM density but intact functional cores. Wet-lab experiments were conducted to validate the designed proteins.

**Strengths:**

1. A novel self-planning algorithm is introduced for diffusion language modeling, enabling the generation of realistic membrane-like protein sequences.
2. Per-Token Guidance, a new classifier-guided sampling algorithm, is proposed to generate sequences with desired properties.
3. Wet-lab experiments were performed to validate the designed proteins’ properties.

**Weaknesses:**

1. The algorithmic novelty is relatively limited.
2. Writing issue: The term *Per-Token Guidance* appears multiple times with inconsistent capitalization or highlighting.
3. Baseline comparison: The baseline methods used for comparison are relatively limited.

**Questions:**

1. Why is it necessary to include non-canonical amino acids in the vocabulary table?
2. In Equation (8), why use a mixture distribution instead of directly maximizing the classifier score within the diffusion model, as is commonly done in classifier-guided diffusion?

---

### Official Review · Reviewer_mJdd · 2025-10-27

**Soundness:** 2
**Presentation:** 2
**Contribution:** 2
**Rating:** 2
**Confidence:** 4

**Summary:**

This work introduces MemDLM, a masked diffusion model based on the RDM framework, specifically designed for the de novo generation of membrane proteins. The authors fine-tuned a pre-trained EvoFlow checkpoint, which features an extended context of 3070 tokens, on a custom-curated dataset. This dataset comprises 11,908 high-quality membrane protein structures (X-ray/EM with < 3.5 Å resolution) sourced from established databases like PDBTM, mpstruc, OPM, and MemProtMD.

For the generation process, the model employs a self-planning sampling scheme, analogous to confidence sampling, to guide the diffusion trajectory. To further enhance the accuracy of amino acid placement at the protein-membrane interface, the authors propose a novel PET guidance strategy.

The model's performance was validated through a series of computational benchmarks. These included assessing the quality of unconditional generation and its efficacy in motif-scaffolding tasks. The scaffolding evaluation was performed on both transmembrane (TM) and soluble motifs, demonstrating the model's versatility in handling different structural contexts within membrane proteins.

Furthermore, the authors conducted a compelling wet-lab experiment to validate their computational results. They used MemDLM to generate several novel single-pass transmembrane (TM) proteins. These de novo proteins were then experimentally tested to determine if they successfully insert into bacterial membranes and exhibit behavior characteristic of naturally occurring TM proteins.

**Strengths:**

The work successfully adapts and applies a diffusion-based framework to the challenge of de novo membrane protein design, addressing a notable gap for recent generative models. Crucially, this computational contribution is substantiated by experimental validation, as the authors provide wet-lab results demonstrating that their generated single-pass proteins can successfully insert into bacterial membranes.

**Weaknesses:**

While the paper presents promising results and valuable experimental validation, the clarity and justification of its methodological contributions could be significantly improved. Several key details and experiments are missing, which makes it difficult to fully assess the novelty and effectiveness of the proposed techniques. The primary weaknesses are detailed below.

#### 1. Lack of methodological clarity and novelty

The paper's core contributions are not clearly distinguished from prior work, and key architectural details are ambiguous.

* Methodological ambiguity: There is a discrepancy in the description of the model. The authors state it is based on the RDM framework but proceed to describe what appears to be a standard masked diffusion process. The precise relationship between these elements needs clarification.

#### 2. Insufficient justification for the PET guidance mechanism

The central methodological contribution, PET guidance, is not sufficiently validated, leaving its effectiveness and complexity unjustified.

* Missing classifier details: A critical omission is the lack of information about the per-token solubility classifier that underpins PET. The paper provides no details on its training data or final performance (e.g., accuracy). Without this, the reliability of the entire guidance mechanism is questionable.
* Absence of ablation studies: The PET mechanism includes several components, such as a prior distribution and "context-aware silency." However, the paper lacks ablation studies to demonstrate the individual contribution of these components. This makes it unclear if the proposed complexity is necessary.
* Lack of comparison to simpler baselines: The authors do not compare PET against simpler or more established guidance techniques (some of which are mentioned in their own Appendix A.2). This makes it impossible to assess whether this new method offers tangible benefits over existing approaches.
* Inconclusive scaffolding experiment: The experiment intended to demonstrate PET's utility ("TOKEN-LEVEL DISCRETE DIFFUSION GUIDANCE") is inconclusive. To properly isolate the effect of the guidance, a crucial baseline is missing: the same model performing scaffold conditioning without PET. Without this direct comparison, the added value of the guidance mechanism remains unproven.

#### 3. Outdated positioning relative to state-of-the-art

The paper's claims regarding its performance are weakened by an outdated view of the current literature.

* The authors refer to EvoDiff as a state-of-the-art (SOTA) discrete diffusion model. However, several more recent and higher-performing models have since been published (e.g., DPLM by Wang et al., DiMA by Meshchaninov et al., Multiflow by Campbell et al.). This incomplete literature review makes it difficult to properly contextualize the performance of MemDLM.

**Questions:**

1.  Could you please clarify the precise relationship between the RDM framework and the masked diffusion process described in your methodology?
2.  The "self-planning" sampling scheme appears functionally identical to the confidence sampling used in models like LLADA (Nie et al., 2025). Could you elaborate on the distinction?
3.  For the scaffolding evaluation, why was the BLOSUM score calculated over the entire protein instead of only the newly generated regions, which would more directly measure the quality of the scaffold?
4.  In Formula 1, should the term be $q(x_t \mid x_0)$?
5.  On lines 195 and 301, the prior distribution is defined using the "previous" timestep $x_{t-1}$. During generation (the reverse process), the previous step is $x_{t+1}$. Could you clarify this notation?

---

### Official Review · Reviewer_2AiL · 2025-11-03

**Soundness:** 1
**Presentation:** 2
**Contribution:** 2
**Rating:** 2
**Confidence:** 4

**Summary:**

The authors address the challenge of designing membrane proteins, which constitute only approximately 1% of known protein structures. They fine-tune EvoFlow, an existing discrete diffusion model based on the Reparameterized Diffusion Model framework, on approximately 12,000 curated membrane protein sequences to create MemDLM. To enable controllable generation, they introduce Per-Token Guidance (PET), a classifier-guided sampling algorithm that selectively modifies editable residue positions while preserving structurally critical transmembrane domains. The approach builds on LaMBO-2's saliency-based guidance by incorporating attention-weighted neighborhoods and mixture distributions. They evaluate generated sequences using computational metrics (pseudo-perplexity, pLDDT, transmembrane residue density) and test five designs experimentally using TOXCAT β-lactamase growth assays in E. coli.

**Strengths:**

1. **The problem is well-motivated and addresses an important gap.**
Membrane proteins constitute only approximately 1% of known protein structures despite their therapeutic importance, and the authors clearly articulate why this creates challenges for computational design methods. The motivation for developing generative models that can handle the unique characteristics of membrane proteins (interleaved transmembrane and soluble regions) is sound and represents a genuine need in the field.

2. **The work includes experimental validation.**
The inclusion of wet-lab experiments using TOXCAT β-lactamase assays demonstrates a commitment to validating computational predictions with experimental data. While the number of tested sequences is limited, the integration of computational design with experimental characterization represents the appropriate direction for advancing protein design methods. The experimental protocols are described in sufficient detail to understand the validation approach.

3. **Dataset curation required substantial effort across multiple sources.**
The authors integrated membrane protein sequences from multiple databases (PDBTM, OPM, mpstruc, MemProtMD) with careful filtering based on resolution and redundancy criteria. While the resulting dataset is modest in size, the curation process involved nontrivial work to identify and annotate transmembrane regions across diverse sources. Sharing this dataset will benefit the broad community.

4. **The goal of controllable membrane protein generation is valuable.**
The motivation for developing methods that can selectively modify proteins while preserving critical structural elements addresses a relevant problem in protein engineering. The general concept of token-level control in generative models for proteins could be useful for various design tasks, even if the specific implementation requires further development or clarification.

5. **Multiple computational evaluation metrics are employed.**
The authors evaluate their generated sequences using several complementary metrics including pLDDT for structural plausibility, pseudo-perplexity for sequence quality, and Shannon entropy for diversity.

**Weaknesses:**

1. **The PET algorithm relies on unjustified assumptions about attention matrices.**
The authors extract "the attention matrix" from the final transformer layer and claim it captures long-range residue information, yet provide no empirical validation of this assumption. With multi-head attention, the method for obtaining a single matrix (averaging across heads, selecting one head, or another approach) is not specified. More critically, the authors do not demonstrate that the attention matrix they use actually encodes the long-range dependencies they claim are necessary for their approach. Also, the algorithm introduces multiple arbitrary hyperparameters without ablation studies to justify these choices or demonstrate their impact on performance.


2. **Section 3.4 (solubilization) lacks baseline comparisons and clear motivation.**
The authors present no comparisons between PET and alternative methods such as ProteinMPNN (e.g. https://doi.org/10.1101/2024.01.16.575764) or fine-tuned versions of EvoDiff or DPLM. The task itself appears poorly motivated: the rationale for solubilizing membrane proteins while preserving TM domains is not clearly connected to therapeutic applications. Furthermore, the authors claim to improve solubility but evaluate only their custom TMRD metric without validating that this metric correlates with actual protein solubility. This disconnect between the stated objective and the evaluation metric undermines the conclusions drawn from this experiment.


3. **The TMRD metric design appears tailored to the results rather than grounded in biological principles.**
* The authors exclude beta-barrel membrane residues from the metric, citing "structural and biophysical differences" between beta-barrels and alpha-helices, but provide no justification for why alpha-helical optimization represents an appropriate scope for membrane protein design. This choice appears to be a post-hoc decision to improve reported performance rather than a principled design based on biological considerations. The authors should either explicitly limit the scope of their entire work to alpha-helical TM proteins or provide substantial justification for this metric construction.
* More fundamentally, TMRD is not a principled metric but rather relies entirely on predictions from DeepTMHMM, an external neural network classifier, as an oracle annotator. The authors provide no justification for why DeepTMHMM predictions should be considered ground truth, nor do they validate that DeepTMHMM performs accurately on their generated sequences, which may be out-of-distribution relative to DeepTMHMM's training data. The paper offers no explanation for why this particular approach to quantifying membrane character should be reliable or what biophysical properties it actually captures.


4. **The baseline comparisons are inconsistent and lack contemporary models.**
The authors select EvoFlow as a base model over DPLM citing its longer context window, yet subsequently compare their results primarily against DPLM rather without establishing an EvoFlow baseline. Sequences, presented in B.6.1 do not require large context, so some clarifications are necessary. ProGen2 evaluations are compromised by the exclusion of one-third of generated sequences due to tokenization issues, preventing fair comparison. There are dozens of available sequence-based models that can be used as baselines in unconditional generation. Also, the authors claim EvoDiff represents state-of-the-art performance (line 415), which is factually incorrect given that DPLM and other more recent models demonstrate comparable superior motif-scaffolding capabilities. This mischaracterization of the field undermines the positioning of the contribution. Again, there are many models that are capable of solving motif-scaffolding/text infilling tasks to evaluate against. The lack of baseline comparisons in Section 3.4 was already mentioned.


5. **MemDLM demonstrates only marginal improvements over existing baselines.**
The performance advantages over DPLM are minimal and inconsistent across evaluation metrics. Generated sequences exhibit substantially lower structural confidence than natural membrane proteins (pLDDT of 67.4 versus 76.6 for the test set). Many reported differences appear to fall within expected variation. For a paper positioning itself as outperforming state-of-the-art methods, the quantitative improvements do not provide strong support for this claim.


6. **Critical analyses are absent throughout the evaluation.**
The paper lacks assessment of sequence diversity or novelty in the generated outputs. The authors do not investigate what distinguishes successful designs from unsuccessful ones beyond the selection criteria applied. No error analysis or systematic examination of failure modes is presented. BLOSUM scores are reported for generated sequences but not for the original dataset, and the metric is inherently inflated by diagonal substitutions (x→x) yielding positive scores. The solubilized protein structures in Figure D.2 appear to contain structural artifacts, and the pLDDT labels do not seem to correspond to the apparent quality of the visualized structures.


7. **Essential details for reproducibility are missing or unclear.**
The motif scaffolding experiments employ a formulation that differs from prior work, yet the authors do not explicitly specify which regions were masked in which proteins, precluding exact reproduction.


8. **The training dataset is modest in size and inadequately characterized.**
Approximately 12,000 sequences represents a relatively small dataset for training deep generative models. The authors motivate their work by highlighting the scarcity of membrane protein structures, yet face this same fundamental limitation in their training data. While various filtering and curation steps are described, no analysis of the resulting dataset is provided. Without summary statistics, diversity metrics, or structural distributions, it is difficult to assess dataset quality or potential biases that may affect model performance. Also, line 888 states that sequences and scripts are "included in SI," but no supplementary files accompany the submission.


9. **The experimental validation, while valuable, is insufficient to support broad claims.**
The inclusion of wet-lab validation is commendable and represents a strength of the work. However, testing only 5 sequences (3 classified as "good" and 2 as "poor") provides limited evidence for the generalizability of the approach. Critically, there is no comparison to the nearest known proteins in existing databases. Without this analysis, it remains unclear whether the generated TM domains represent genuinely novel designs or closely resemble existing sequences, which would suggest memorization rather than de novo design capability.


10. **The experimental results contain ambiguities that question the selection framework.**
PoorTM2 exhibited growth in carbenicillin despite being classified among the bottom 22 sequences by computational metrics. This observation directly contradicts the premise that the model's computational scores effectively distinguish high-quality from low-quality TM sequences. If a sequence ranked in the bottom 2% can successfully insert into membranes, the relationship between computational predictions and experimental function requires clarification.


11. **The claimed novelty is overstated relative to the methodological contributions.**
Describing this as the "first experimentally-validated diffusion-based model for membrane protein generation" based on 5 tested sequences represents a significant overstatement. Claims to "outperform state-of-the-art" are not consistently supported across metrics and baselines. MemDLM itself constitutes fine-tuning of an existing model (EvoFlow) on domain-specific data. PET adapts LaMBO-2's approach with modifications for token preservation. The primary contribution is the application of existing methods to membrane protein design rather than fundamental methodological innovation.

12. **Reference formatting does not conform to the submission guidelines and requires revision.** E.g. lines 050 and 207.

**Questions:**

1. How is the single attention matrix extracted from multi-head attention? Is it averaged across heads, or is a specific head selected?
2. What is the computational cost of PET sampling compared to standard P2 sampling, and how many optimization steps are typically required?
3. What is the sequence similarity between the generated/evaluated TM domains and the nearest known sequences in existing databases (PDB, UniProt etc.)?
4. How does the per-token solubility classifier (vφ) perform on generated sequences compared to natural membrane proteins? What is its accuracy on held-out test data, and how does it handle sequences that may be out-of-distribution?
5. How sensitive are the PET results to the various hyperparameters (γ=0.5, α=5.0, temperature τ, top-K selection)?
6. Are all of the heuristics from the Section 2.3 required for the model to work? Can you provide the ablations that would clarify, if all of these seemingly unjustified decisions really add up to the model performance?
7. How does PET compare to simpler baseline guidance methods, such as directly masking conserved positions without the attention-based neighborhood construction or the mixture distribution formulation?

---

### Official Review · Reviewer_6DfY · 2025-11-05

**Soundness:** 3
**Presentation:** 2
**Contribution:** 2
**Rating:** 4
**Confidence:** 4

**Summary:**

This paper addresses the challenge of membrane protein design, which is hindered by scarce high-resolution structures and suboptimal energy functions, by introducing MemDLM, a discrete diffusion-based protein language model fine-tuned on membrane protein sequences.
Built on the reparameterized diffusion model (RDM) framework and EvoFlow backbone, MemDLM enables de novo generation, motif scaffolding, and controllable solubilization of membrane proteins. To enhance controllability, the authors propose per-token guidance (PET), a classifier-guided sampling strategy that preserves conserved transmembrane (TM) domains while solubilizing non-critical residues. MemDLM outperforms baselines (DPLM, ProGen2, EvoDiff) on computational metrics (pLDDT, TM residue density, perplexity) and is experimentally validated via TOXCAT-β-lactamase assays, where its "GoodTM" designs exhibit successful membrane insertion. The work positions MemDLM as the first experimentally validated diffusion model for rational membrane protein design.

**Strengths:**

1. PET’s integration of attention-weighted neighborhoods and saliency scores for token-level control is a creative combination of classifier guidance and structural biology insights, addressing the unique need to preserve TM domains.
2. Experiments are solid, with both computational benchmarks (covering generation, scaffolding, solubilization) and wet-lab validation (TOXCAT assays) that confirm biological functionality, not just structural plausibility.
3. MemDLM has the potential to fill a critical gap in membrane protein design by enabling end-to-end, controllable generation without relying on pre-defined scaffolds, with potential implications for therapeutics and synthetic biology.

**Weaknesses:**

- As shown in section B.6.1, the TOXCAT assay validates only 5 designs (3 GoodTM, 2 PoorTM) with short sequences (~20–30 residues), focusing on single-pass TM helices. No validation of larger, multi-pass membrane proteins (common in biology) or functional assays (e.g., ligand binding, signaling) is provided, leaving MemDLM’s utility for real-world therapeutic design unproven.
- MemDLM is a fine-tuned variant of EvoFlow (an existing RDM-based model), and its training objective is directly adopted from prior work (Zheng et al., 2023; Wang et al., 2024). The only novel component, the PET builds heavily on LaMBO-2 (Gruver et al., 2024) for discrete classifier guidance, with incremental, yet meaningful, changes (min-max saliency scaling, attention neighborhoods), which also lack a compelling theoretical and empirical justification for superiority over other classifier-guided approaches for dLLMs.
- PET requires a pre-trained solubility classifier (ESM-2-650M-based) and additional inference steps, but the paper does not report latency or computational cost, which is critical for deployment. There is no ablation of PET’s components (e.g., neighborhood attention, saliency scaling) to show which drive performance.
- How does MemDLM perform on membrane proteins with non-α-helical folds (e.g., β-barrels)? Your training data seeded dominated by α-helices, so generalizability to other membrane topologies is unclear.

**Questions:**

see weaknesses.

---

### Note · Authors · 2025-11-14

**Comment:**

We thank the reviewers and the area chair for taking the time to evaluate our submission, **“Token-Level Guided Discrete Diffusion for Membrane Protein Design.”** We appreciate the effort that goes into a full set of conference reviews. At the same time, we want to be honest about our reaction to the feedback. Several aspects of the reviews were surprising and, in many cases, difficult to reconcile with the practical realities of an ICLR revision cycle.

A large fraction of the weaknesses raised across all reviewers focus on requests for significant new wet lab experiments. These include multi pass membrane proteins, expanded functional assays, detailed classifier testing on out of distribution constructs, broad ablations that depend on new biological measurements, and new experimental controls for membrane insertion and signaling. Many of these suggestions would require new cloning and expression work, new bacterial selection experiments, and new rounds of assay optimization. These tasks cannot be completed in two weeks. They cannot even be completed in one month. They require substantial laboratory time and resources.

It is disappointing that the reviews frame these large scale experimental expansions as if they were expected for a conference paper. Even the reviewers who rated the paper positively request biological studies that go far beyond what any group could produce within the revision window. We also note that several methodological comments ask for clarifications that are either already present in the submission or based on misunderstandings about how discrete diffusion models work. This makes it difficult to know what form of revision would actually satisfy the reviewers, even if we set aside the wet lab timeline problem.

We are also disappointed that the experimental validation that was provided, which included cloning and full TOXCAT beta lactamase growth assays, was often minimized or treated as insufficient. These assays clearly separated successful designs from unsuccessful ones. They also demonstrated that the model is capable of producing **de novo** sequences that insert into bacterial membranes. For a conference paper, this level of validation is already uncommon. It is surprising that the reviews treat this as a small or expected contribution, while simultaneously requesting experiments that would take months of additional work.

Given these issues, it is clear that a meaningful revision is not possible within the constraints of an ICLR cycle. Completing the requested experiments would require new design cycles, new cloning, new expression work, and new biological analysis. We cannot and should not attempt to rush any of these steps. Doing so would compromise the work and would not meet the standards of our own experimental program.

For these reasons, we are withdrawing the submission. We will expand the biological validation, address the suggested ablations with proper data, and prepare a complete journal version where the work can be evaluated with the appropriate expectations for both computational and experimental strengths.

We thank the area chair and the reviewers for their time, but we cannot pursue a revision path under the current expectations.

**Withdrawal Confirmation:**

I have read and agree with the venue's withdrawal policy on behalf of myself and my co-authors.